# SLoPe: Double-Pruned Sparse Plus Lazy Low-Rank Adapter Pretraining of LLMs

**Mohammad Mozaffari**
Department of Compute Science
University of Toronto
mmozaffari@cs.toronto.edu

**Amir Yazdanbakhsh**
Google DeepMind
Mountain View, USA
ayazdan@google.com

**Zhao Zhang**
Department of Electrical and Computer Engineering
Rutgers University
zhao.zhang@rutgers.edu

**Maryam Mehri Dehnavi**
Department of Compute Science
University of Toronto
mmehride@cs.toronto.edu

## Abstract

We propose SLoPe, a Double-Pruned **S**parse Plus Lazy **Lo**w-rank Adapter **Pre**training method for LLMs that improves the accuracy of sparse LLMs while accelerating their pretraining and inference and reducing their memory footprint. Sparse pretraining of LLMs reduces the accuracy of the model, to overcome this, prior work uses dense models during fine-tuning. SLoPe improves the accuracy of sparsely pretrained models by adding low-rank adapters in the final 1% iterations of pretraining without adding significant overheads to the model pretraining and inference. In addition, SLoPe uses a double-pruned backward pass formulation that prunes the transposed weight matrix using N:M sparsity structures to enable an accelerated sparse backward pass. SLoPe accelerates the training and inference of models with billions of parameters up to $1.25\times$ and $1.54\times$ respectively (OPT-33B and OPT-66B) while reducing their memory usage by up to $0.63\times$ and $0.61\times$ for training and inference respectively.[1]

## 1 Introduction

Large Language Models (LLMs) demonstrate significant potential for natural language understanding and generation; however, they are expensive to train and execute because of their extensive parameter count and the substantial volume of training data required. The training process of LLMs include a pretraining (45) and a fine-tuning stage. In the pretraining phase, the model is trained on a large high-quality text (17; 1) and then fine-tuned on different downstream tasks (57; 48). Both phases require significant amounts of computation, memory, and communication.

Model sparsity, in which the less important parts of the model are pruned, can reduce the computation and memory overheads of LLM pretraining (24). Sparsity is unstructured if elements are removed from arbitrary locations in the tensors. Unstructured sparsity is hard to accelerate due to non-existing hardware/software support (58). To resolve this, structured sparsity imposes constraints on where the zero elements can appear (28; 33), creating dense blocks of nonzeros in the matrix to leverage dense compute routines. The drawback of the structured sparse methods is that they limit the choice for sparsity patterns leading to a reduction in accuracy in the sparse model when compared to dense (9). NVIDIA has recently introduced sparse tensor cores (43) to their hardware that accelerate more flexible structured sparsity patterns, i.e. 2:4 sparsity; hardware support for N:M sparsity where at most N out of M consecutive elements are zero is not yet available but machine learning practitioners are developing algorithms for these patterns (29; 34; 47) .

Applying N:M sparse masks to a model leads to accuracy loss because of their limited choice of sparsity patterns. Changing the sparsity mask dynamically throughout pretraining is one of the approaches proposed to address this issue (11). Zhou et al. (61) proposes a novel metric for finding the N:M sparsity patterns that lead to higher accuracy in each iteration. (29) suggest the use of decaying masks to further improve the accuracy of the models. STEP (34) proposes a new optimizer

---

[1] Code and data for SLoPe is available at: https://bit.ly/slope-llm

that improves the convergence of models with adaptive masks. While the adaptive methods can improve the accuracy of the models, they require storing the dense weights and possibly additional metrics for updating the new sparsity patterns, while wasting a portion of the training computations to train the weights that will be pruned in later iterations. SPDF (55) and Sparse-Dense Pretraining (FST) (26), one can compensate for the loss imposed by sparsity with a dense fine-tuning. But the dense fine-tuning stage will disable the memory and compute savings of sparse methods at inference. Inspired by this, we introduce additional non-zeros to the weight in the last steps of pretraining. To avoid storing a dense model during inference while getting the same capabilities of a dense weight, we add the non-zeros in the form of low-rank adapters (25). Our experiments show that using low rank adaptors leads to noticeably faster convergence compared to when the same number of learnable parameters are added to the sparse weights.

The use of N:M sparsity in LLM pretraining is limited to accelerating the forward pass in the training loop because the row-wise N:M structure in the weight sparsity pattern will be lost when the weights are transposed in the backward pass. Prior work (27; 60; 26) attempt to leverage sparsity in both forward and backward passes by finding transposable masks through various methods: greedy search algorithms, searching among random permutations, and searching among the results of convolution. However, these transposable masks reduce model accuracy and add significant runtime overheads (26), often resulting to slow-downs (up to $8.4\times$). To address these issues, we propose a **double-pruned backward pass** formulation with theoretical convergence guarantees. Instead of enforcing the weight transpose to be N:M sparse, our approach transposes the N:M weight matrix first and then imposes N:M sparsity. This allows the weight matrices to exhibit a wider range of sparsity patterns, leading to improved accuracy.

Our method, SLoPe, is a Double-Pruned **S**parse Plus Lazy **Lo**w-rank Adapter **Pre**training method for LLMs. It employs a *static* N:M sparsity mask with a double-pruned backward pass formulation to accelerate both the forward and backward passes. Key contributions of SLoPe are:

- **Double-Pruned backward pass** $\rightarrow$ We propose to transpose an already sparsified N:M weight matrix (forward pass) before imposing another round of N:M sparsity (backward pass), improving model quality and reducing mask search overheads.

- **Lazy Low-Rank adapters** $\rightarrow$ We introduce additional parameters with minimal compute and memory overheads, merely for the last 1% iterations of pretraining, improving model capacity (see Figure 1).

- **Optimized CUDA kernels** $\rightarrow$ We jointly optimize Nvidia 2:4 sparse kernels and low-rank calls through efficient tiling and scheduling. Our highly-optimized CUDA kernels result to $1.25\times$ end-to-end training speedup and $1.54\times$ inference speedup on LLMs with billions of parameters, while reducing training and inference memory footprint by up to $0.63\times$ and $0.61\times$, respectively.

## 2   SPARSE PLUS LOW-RANK PRETRAINING OF LLMS

Equation 1, 2, and 3 depict the formulas for the forward and backward pass of the $i$-th linear layer in a neural network. Here, the weight tensor is denoted as $\mathcal{W}_i \in \mathbb{R}^{d_{out} \times d_{in}}$ and the input tensor is denoted as $\mathcal{X}_i \in \mathbb{R}^{b \times d_{in}}$. The forward pass generates an output tensor represented as $\mathcal{Y}_i \in \mathbb{R}^{b \times d_{out}}$. In all equations, $d_{in}$ and $d_{out}$ refer to the input and output dimensions of the respective layer.

$$\text{FWD} \mapsto \mathcal{Y}_i = \mathcal{X}_i \mathcal{W}_i^T \tag{1}$$

$$\text{BWD} - 1 \mapsto \nabla_{W_i}\mathcal{L} = \nabla_{Y_i}\mathcal{L}^T \mathcal{X}_i \tag{2}$$

$$\text{BWD} - 2 \mapsto \nabla_{X_i}\mathcal{L} = \nabla_{Y_i}\mathcal{L} \mathcal{W}_i \tag{3}$$

The dimension along which N:M pruning occurs corresponds to the reduction dimension in Matrix-Matrix multiplication. Without this restriction, the sparse Matrix-Matrix operation can not be accelerated on GPU (41). With this restriction in mind, to leverage weight sparsity in forward and backward pass, one needs to prune elements along the columns of $\mathcal{W}_i^T$ in Equation 1 (FWD) and $\mathcal{W}_i$ in Equation 3. To satisfy this requirement, it is necessary to prune elements of the weight tensor $\mathcal{W}_i$ along both row and column dimensions.

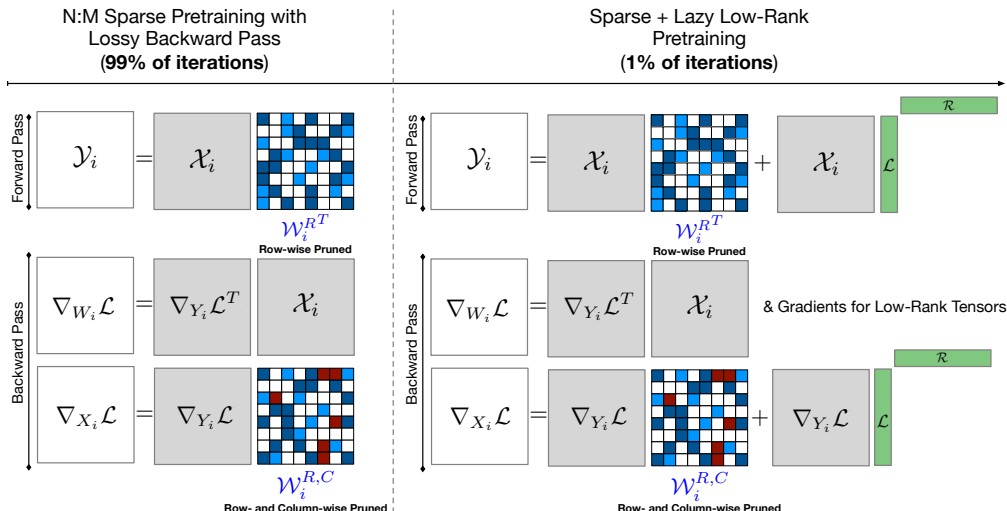

Figure 1: The sparse training pipeline in SLOPE. Here, $\mathcal{X}$, $\mathcal{Y}$, and $\mathcal{W}$ denote the input, output, and the weight tensors for a specific layer, respectively. $\nabla_.\mathcal{L}$ represents the gradient of the loss function. $\mathcal{L}$ and $\mathcal{R}$ are the low-rank terms that are introduced only in the final 1% iterations. Superscript $R$ shows row-wise pruning using $N{:}M$ scheme and $R, C$ shows both column and row-wise $N{:}M$ sparsification, leading to extra imposed zeros. Blue elements represent non-zero values, while white elements represent pruned values, and red elements indicate additional zeros introduced during the backward pass.

## 2.1 DOUBLE-PRUNED BACKWARD PASS

Various approaches can be used to exploit N:M sparsity during both the forward and backward passes. For example, one may prune the activation tensor $\mathcal{X}_i$ in FWD along the row dimension and $\mathcal{W}_i$ in BWD-2 along the column dimension. Although diverse combinations exist for pruning, our focus in this study is primarily on the sparsification of weight tensors for two reasons: (a) the sparsification of weight tensors directly impact the resource required for model storage and serving, and (b) our initial findings indicate that pruning weight tensors during both forward and backward passes has a comparatively lesser adverse impact on the overall end-to-end model quality. More details on our experiments can be found in J. As such, we posit a *double-pruned backward pass* formulation that can productively accelerate FWD and BWD-2 computations.

In addition, we prove that such materialization of pruned weight tensors, despite being lossy[2], exhibits convergence properties. For the rest of this paper, we represent the weight tensor subjected to row-wise pruning as $\mathcal{W}_i^R$, while the concurrent row-wise and column-wise pruning (double-pruned) is presented as $\mathcal{W}_i^{R,C}$. We rewrite the training equations to accommodate these modifications, with proposed changes highlighted in blue:

$$\text{FWD} \mapsto \mathcal{Y}_i = \mathcal{X}_i \mathcal{W}_i^{R^T} \tag{4}$$

$$\text{BWD} - 1 \mapsto \nabla_{W_i}\mathcal{L} = \nabla_{Y_i}\mathcal{L}^T \mathcal{X}_i \tag{5}$$

$$\text{BWD} - 2 \mapsto \nabla_{X_i}\mathcal{L} = \nabla_{Y_i}\mathcal{L}\mathcal{W}_i^{R,C} \tag{6}$$

Using this formulation for training, we can accelerate both forward and backward passes owing to the existence of N:M sparsity along both dimensions of weight tensors.

**Memory footprint analysis.** Inducing N:M structured sparsity not only improves computational efficiency of GEMM operations but also reduces the memory footprint for storing sparse tensors.

---

[2]We term this formulation "*lossy*" because the weight matrix undergoes information loss during the backward pass compare to its state in the forward pass.

It is noteworthy, however, that the storage of auxiliary meta-data becomes necessary, containing information about the locations of non-zero elements in a supporting matrix. Equation 7 delineates the requisite number of bits for storing the indices in the N:M sparsity format, where $\lceil . \rceil$ denoting the ceiling function. We present the detailed results on the memory footprint reduction in section 3.

$$n_{index}^{N:M} = \left\lceil log\left(\binom{M}{N}\right) \right\rceil \tag{7}$$

**Convergence analysis.** Theorem 2.1 (proof in subsection T.1) shows the additional sparsity resulting from double pruning to an initially row-wise N:M pruned matrix. Following this lemma, we quantify the increased sparsity induced by double pruning with 1:2, 2:4, and 2:8 sparsity patterns as $12.5\%$, $9.375\%$, and $3.39\%$, respectively. This observation underscores that as the value of M in N:M increases, the surplus of zero elements in a double-pruned matrix diminishes. This reduction in zero elements consequently implies a decrease in computational errors, enhancing the robustness of the computations. We expound further insights into this phenomenon in Appendix I.

**Lemma 2.1.** *Consider a randomly initialized matrix A. Following our notations, we denote the row-wise pruned version of A by $A^R$ and the joint column- and row-wise pruned version of A by $A^{R,C}$. We use $D(.)$ to present the density ratio of a matrix. Equation 8 shows the additional zero elements in matrix A that are introduced by double-pruning, where $s = \frac{N}{M}$.*

$$D(A^R) - D(A^{R,C}) = \sum_{j=N+1}^{M} \binom{M}{j} s^j (1-s)^{M-j} \frac{j-N}{M} \tag{8}$$

Theorem 2.2 states that the dynamic alteration of the column-wise mask in Equation 5 during each training iteration does not exert a detrimental impact on the convergence of the optimizer. This phenomenon can be attributed to the equivalence between the left-hand side of Equation 9, which corresponds to Equation 3 [BWD-2], and the averaging effect achieved through multiple training iterations of backpropagation with distinct sparsity mask. However, for arbitrary values of N and M, 4 and 5 can be used in the training with convergence guarantee (proof in subsection T.1). The sparsity mask is chosen randomly at initialization, i.e. all the weights have the same probability of being zero or non-zero. This is because at initialization the location of weights with larger magnitude is arbitrary. After choosing the sparsity mask at initialization, we keep the mask fixed throughout the entire training process. This policy ensures that each element in the weight has the same probability of being non-zero at initialization and satisfies the assumption in Lemma 2.1.

**Theorem 2.2.** *Assuming a loss function $\mathcal{L}(\mathcal{W}_i, \mathcal{X}_i)$ for a random sample $X_i$, and considering a random mask $M_i$, Equation 9 holds, where $E[.]$ is the expectation operator and $\odot$ is the element-wise multiplication.*

$$E_{X_i}[\nabla_{X_i}\mathcal{L}(W_i, X_i)] = \frac{M}{N} E_{M_i}[E_{X_i}[\nabla_{Y_i}\mathcal{L}(W_i, X_i)(M \odot W_i)]] \tag{9}$$

## 2.2 Lazy low-rank adapters

Pruning weight tensors in FWD and BWD-2 computations is desirable for computational efficiency but may have detrimental impact on quality. To mitigate this adverse impact on model quality, we augment the doubly-pruned weight matrix with a low-rank matrix. The decomposition of the doubly-pruned weight matrix, combined with the low-rank matrix, maintains the computational efficiency of spare Matrix-Matrix multiplication during forward and backward passes. Simultaneously, this approach holds promise in alleviating the adverse effects of double pruning on overall model quality.

Considering the dense weight matrix, denoted by $W_{dense} \in \mathbb{R}^{d_{out} \times d_{in}}$, Equation 10 illustrates the proposed matrix decomposition. In this expression, $W_{sparse} \in \mathbb{R}^{d_{out} \times d_{in}}$ signifies a doubly-pruned matrix and $L \in \mathbb{R}^{d_{out} \times r}$ and $R \in \mathbb{R}^{r \times d_{in}}$ are components of the low-rank approximation. The variable $r$ denotes the rank of this low-rank approximation. $r$ functions as a hyperparameter that controls the trade-offs between memory footprint, computational efficiency, and model quality.

$$\mathcal{W}_{dense} = \mathcal{W}_{sparse} + \mathcal{L}\mathcal{R} \tag{10}$$

The matrix decomposition of doubly-pruned matrix combined with a low-rank matrix approximation reduces the memory footprint of $\mathcal{W}$ from $d_{in}d_{out}$ to $d_{in}d_{out}\frac{N}{M} + (d_{in} + d_{out})r$, where $r <<$ $min(d_{in}, d_{out})$. The computational complexity of dense Matrix-Matrix multiplication, however, changes from $bd_{in}d_{out}$ to $bd_{in}d_{out}\frac{N}{M} + b(d_{in} + d_{out})r$. Given the substantially smaller value of $r$ in comparison to $b$, $d_{in}$, and $d_{out}$, our formulation effectively reduces both memory footprint and computational complexity of Matrix-Matrix multiplication by a factor of $\frac{M}{N}\times$.

We empirically show that the convergence rate of low-rank adapters surpasses that of sparse weights. We attribute this behavior to the notably lower parameter counts inherent in low-rank adapters. Leveraging this observation, we incorporate low-rank adapters exclusively during the final 1% of the training iterations. This confined usages of low-rank adapters results in additional reduction of training cost, specifically in terms of total number of operations. We term the proposed usage of low-rank adapters in the final steps of the training as *lazy low-rank adapters*.

## 2.3 SPARSE KERNELS

cuSPARSELt is a CUDA library designed explicitly for sparse Matrix-Matrix multiplication, where one operand undergoes pruning with the 2:4 sparsity pattern. However, this library does not offer APIs for other algebraic routines such as addition and assignment for sparse tensors. We now delve into the details of different kernels for training and overview our implementation methodology.

Algorithm 1 shows the training process of a single linear layer taken from an attention-based model. We assume the use of weight decay in the optimizers, and subsequently design the requisite sparse APIs to facilitate the optimizer operations. The training starts with matrix initialization (line 2) and setting up sparse formats to store weight tensors and their corresponding transpose (line 3 and 4). Then, for every mini-batch in the training set, we compute the forward pass following Equation 4 (line 8). As part of the backward pass, the derivative of the loss function with respect to the output activation is computed (line 10). Subsequently, the gradients of the loss function with respect to the input activation (line 11) and the weight tensor (line 12) are computed using Equation 5 and Equation 2, respectively. In order to circumvent the necessity of updating weights with zero values and mitigate the associated memory footprint overhead, we employ a strategy wherein we mask the gradients for pruned weights. The computed values are stored in a sparse format (line 13). Next, in order to implement weight decay in the optimizer and mitigate the impact of gradient scaling, we compute the value of $\frac{1}{\gamma}\nabla_W\mathcal{L} + \alpha W$ (line 15). Here, $\alpha$ is the weight decay applied in the optimizer, while $\gamma$ denotes the gradient scaling factor for numerical stability during the half-precision backward pass. The updated values for the weight tensor are calculated according to the optimizer update rule (line 16). Finally, the value of weight tensor and its transpose are updated directly in a sparse format (line 17 and line 18). More details about the implementation of the custom kernels used in Algorithm 1 can be found in Appendix K.

## 2.4 SLoPe RUNTIME OPTIMIZATION

While SLoPe improves the training and inference of LLMs by introducing sparse weights and low-rank adapters, a naïve implementation can hinder its full performance improvement. Specifically, cuSPARSELt (40) SpMM kernels exhibit sensitivity to input and weight tensor shapes, and introducing low-rank adapters at inference increases can increase the number of calls during the forward pass of each linear layer. This section covers our approach to optimize SLoPe's implementation and further improve model performance.

**Efficient tiling of upsample tensors.** Figure 3-**(a)** showcases the speedup achieved by the cuS-PARSELt backend across a range of tensor shapes commonly used in LLMs. While the speedup of SpMM in downsample tensors increases gradually as their sizes increase, the speedup of upsample tensor drops off at around hidden dimension = 4000. To overcome this limitation, we tile the upsample tensor into multiple smaller matrices of equal size, each of which benefits from improved speedup when multiplied by the input using 2:4 sparsity. By tuning the size of the tiles, we figured that the best performance can be achieved by using square tiles. The results of these multiplications are then concatenated. This optimization, as detailed in Appendix E, leads to a `12%` improvement in inference speed and a `4%` increase in training speed with SLoPe.

**Efficient kernel for combined SpMM+low-rank adapters.** A straightforward implementation of low-rank adapters requires four kernel calls: one for sparse matrix multiplication, two for low-

---

**Algorithm 1** Accelerated Sparse Pretraining Algorithm for a Linear Layer

---

1: **Input:** Weight: $W$, Training Set: $\mathbb{D}$, Weight Decay: $\alpha$, Gradient Scaling Factor: $\gamma$
2: backend.init()
3: WSparseTranspose = backend.setup(W.tranpose())
4: WSparse = backend.setup(W)
5: sparseMask = (WSparse != 0) *// Element-wise*
6: **for** $(X, \hat{Y}) \in \mathbb{D}$ **do**
7:     *// Forward Pass*
8:     Y = backend.spmm(X, WSparseTranspose)
9:     *// Backward Pass*
10:    gradOutput = $\nabla_Y \mathcal{L}$
11:    gradInput = backend.spmm(gradOutput, WSparse)
12:    gradWeight = backend.matmul(gradOutput.transpose(), X)
13:    gradWeightSparse = backend.pruneAndCompress(gradWeight, sparseMask)
14:    *// Optimizer with Weight Decay*
15:    g = backend.sparseAdd(gradWeightSparse, WSparse, $\frac{1}{\gamma}$, $\alpha$)
16:    WNew = optimizer.updateWeight(g)
17:    backend.updateSparseMatrix(WSparse, WNew)
18:    backend.updateSparseMatrix(WSparseTranspose, WNew.transpose())
19: **end for**

---

rank computations, and one for adding the results. In addition, our experiments demonstrate that multiplying matrices with low-rank adapters does not scale proportionally with the adapter's rank, leading to significant overheads due to their low arithmetic intensity (see Appendix C). To address this, we introduce two optimizations: *(1)* concatenating the downsample tensor to the sparse weight tensor, reducing kernel calls and increasing arithmetic intensity as in Equation 11-left, and *(2)* leveraging a cuBLAS fused matrix multiplication and addition kernel, minimizing cache access and kernel calls as in Equation 11-right. As demonstrated in Appendix D, these optimizations collectively contribute to a speedup improvement of up to 6% in the end-to-end inference speed.

$$[\mathcal{Y}_1 | \mathcal{Y}_2] = \mathcal{X}[\mathcal{W}^T | \mathcal{L}]; \qquad \mathcal{Y} = \mathcal{Y}_2 \mathcal{R} + \mathcal{Y}_1 \tag{11}$$

## 3 EXPERIMENTAL RESULTS

This section evaluates the efficacy of SLOPE in accelerating the pretraining while achieving memory savings. Due to the substantial computational resources required for LLM pretraining, our accuracy evaluation is primarily focused on smaller-scale LLMs up to 774M parameters. However, the speedup and memory reduction results extend to a wider range of models, from 2.6B up to 66B parameters.

### 3.1 END-TO-END SPEEDUP AND MEMORY SAVING: PRETRAINING AND INFERENCE

We evaluate the speedup and memory reduction by SLOPE during pretraining and inference across LLMs with different model parameter sizes. To demonstrate the scalability and efficiency of our method, we conducted extensive benchmarking on OPT (2.6 B to 66 B) and LLaMA-3-8B and Mistral-v0.3-7B models. In all the experiments, we have enabled FlashAttention-2 (8) (Appendix M presents detailed ablation study on the impact of FlashAttention). To mitigate the impact of outliers, we conducted 1,000 iterations for each speedup experiment and reported the median value. For the memory reduction experiments, we performed five independent runs and similarly reported the median outcome. These methodologies were chosen to provide a more reliable measure of central tendency in our results [3].

---

[3]It is noteworthy that for benchmarking speedup and memory savings, which require comparatively fewer computational resources than comprehensive pretraining accuracy experiments, we utilized the OPT, LLaMMA-3, and Mistral-v0.3 model families. These families were selected due to their diverse range of model parameter sizes, allowing for a more thorough study of performance across different scales.

Table 1: Comparative analysis of end-to-end pretraining and inference speedup ($\times$) comparison between SLoPe and the latest work (FST) on accelerating pretraining with 2:4 sparsity (ICML 2024) (26). Note that the lack of inference speedup in FST is because of the final dense pretraining during the final iterations, resulting in a dense model for inference. E-SR-STE stands for Extened SR-STE.

| MODEL | METHOD | TRAINING No Adapter ($r = 0$) | INFERENCE No Adapter ($r = 0$) | 1.56% Adapter | 6.25% Adapter |
|---|---|---|---|---|---|
| OPT-66B | SLoPe | **1.20** | **1.46** | **1.43** | **1.40** |
|  | FST | 1.06 | 1.00 | 1.00 | 1.00 |
| OPT-30B | SLoPe | **1.22** | **1.53** | **1.53** | **1.50** |
|  | FST | 1.07 | 1.00 | 1.00 | 1.00 |
| OPT-13B | SLoPe | **1.25** | **1.54** | **1.39** | **1.36** |
|  | FST | 1.10 | 1.00 | 1.00 | 1.00 |
| OPT-6.6B | SLoPe | **1.21** | **1.46** | **1.46** | **1.43** |
|  | FST | 1.11 | 1.00 | 1.00 | 1.00 |
| OPT-2.6B | SLoPe | **1.13** | **1.31** | **1.25** | **1.18** |
|  | FST | 1.09 | 1.00 | 1.00 | 1.00 |
| LLaMA-3-8B | SLoPe | **1.16** | **1.35** | **1.33** | **1.32** |
|  | FST | 1.09 | 1.00 | 1.00 | 1.00 |
| Mistral-v0.3-7B | SLoPe | **1.15** | **1.34** | **1.32** | **1.31** |
|  | FST | 1.07 | 1.00 | 1.00 | 1.00 |

We compared our method against dense pretraining and inference directly in PyTorch, which uses efficient cuBLAS backend. As the sparse pretraining benchmark, we compare our work against Sparse-Dense Pretraining (FST) (26), the state-of-the-art 2:4 pretraining method and the only semi-structured sparse pretraining work that provides end-to-end speedups. Note that methods targeting LLM pretraining with N:M sparsity often suffer from inefficiency due to mask search overheads and/or compression setup. Appendix H and Appendix B detail the profiling in Bi-Mask (60) and FST (26), which similarly use N:M sparsity on both forward and backward passes.

Notably, our approach, SLoPe, diverges significantly from recent work Fully Sparse Training (FST) (26) in two key aspects. Firstly, *we comprehensively prune all weights in the model, encompassing both MLP and Self-Attention modules*, whereas FST only prunes weights in the MLP modules. Secondly, FST employs dynamic transposable weights, which introduce additional computation and memory overhead during training. Finally, FST necessitates dense fine-tuning ($\sim$17% of pretraining), thereby negating their speedup advantages during inference. In contrast, our approach achieves efficient and accurate large language models during both training and inference without such limitations.

**SLoPe speedup for pretraining and inference.** Table 1 summarizes the speedups achieved by our method during both training and inference. Since over 99% of training occurs without low-rank adapters, the training speedup is largely independent of the adapter rank. Conversely, inference speedup is directly influenced by the adapter rank. Given the varying hidden dimensions across different model sizes, we report the inference speedup for various adapter rank ratios: $\frac{adapter-rank}{hidden-dimension}$.

Figure 3-**(a)** illustrates that cuSPARSELt achieves higher speedups for large matrices until it reaches its maximum performance capacity ($2\times$). A similar trend is observed in the pretraining and inference speedups of the models. For small matrices used in low-rank adapters, the lower arithmetic intensity of low-rank adapter multiplication results in higher overhead relative to sparse multiplication. This is because low arithmetic intensity limits the full utilization of GPU resources, leading to inefficiencies.

**SLoPe memory reduction in pretraining and inference.** For training, the memory consumption of a dense model includes weights, gradients, and optimizer states, amounting to $4 \times 16$ bits for weights, $4 \times 16$ bits for gradients, and $2 \times 4 \times 32$ bits for optimizer states. The sparse model, however, stores non-zero weights and indices twice (for both weights and transposed weights), along with a binary mask, gradients, and reduced optimizer states. This adds up to $2 \times (16 + 3)$ bits (weights and transposed weights), $4 \times 8$ bits (binary mask), $2 \times 16$ bits (gradients), and $2 \times 2 \times 32$ bits (optimizer states). Consequently, the memory footprint during training is reduced by 68%. For inference, a dense model requires storing weights with a total memory cost of $4 \times 16$ bits. In contrast, our sparse

Table 2: Comparative analysis of end-to-end memory reductions ($\times$) during training and inference between SLoPe and the latest work (FST) on accelerating pretraining with 2:4 sparsity (ICML 2024) (26). Values greater than $1.00\times$ show memory overhead.

| MODEL | METHOD | TRAINING No Adapter ($r = 0$) | INFERENCE No Adapter ($r = 0$) | 1.56% Adapter | 6.25% Adapter |
|---|---|---|---|---|---|
| OPT-66B | SLoPe | **0.67** | **0.63** | **0.65** | **0.70** |
| | FST | 1.27 | 1.00 | 1.00 | 1.00 |
| OPT-30B | SLoPe | **0.67** | **0.61** | **0.63** | **0.69** |
| | FST | 1.17 | 1.00 | 1.00 | 1.00 |
| OPT-13B | SLoPe | **0.68** | **0.51** | **0.62** | **0.68** |
| | FST | 1.16 | 1.00 | 1.00 | 1.00 |
| OPT-6.6B | SLoPe | **0.68** | **0.60** | **0.62** | **0.68** |
| | FST | 1.19 | 1.00 | 1.00 | 1.00 |
| OPT-2.6B | SLoPe | **0.67** | **0.62** | **0.64** | **0.70** |
| | FST | 1.18 | 1.00 | 1.00 | 1.00 |
| LLaMA-3-8B | SLoPe | **0.63** | **0.66** | **0.69** | **0.71** |
| | FST | 1.17 | 1.00 | 1.00 | 1.00 |
| MISTRAL-v0.3-7B | SLoPe | **0.68** | **0.66** | **0.69** | **0.65** |
| | FST | 1.15 | 1.00 | 1.00 | 1.00 |

model optimizes memory usage by storing only the non-zero weights and their indices, resulting in $2 \times 16$ bits for non-zeros and three bits for indices (see equation 7). This leads to a 54% reduction in memory usage during inference.

Table 2 presents the memory reduction for different low-rank adapter ranks and OPT, LLaMA-2, and Mistral model variants. The memory reduction is slightly less than the theoretical expectation, primarily because of additional memory usage from other model components, such as layer norms, and dense model parameters.

## 3.2 PRETRAINING ACCURACY RESULTS

To assess the impact of SLoPe on model accuracy, we conducted pretraining experiments across various models and datasets (details in Appendix O). In all experiments, the classifications heads and the first linear layer following the input are dense.

**GPT2 (Small/Large).** We pretrained both the small (117 M parameters) and large (774 M parameters) variants of GPT2 (46) on the OpenWebText dataset (1). For a fair comparison, we evaluate the models on MMLU (23), Arc Challenge (6), and OpenBookQA (35) zero-shot tasks implemented in Language Model Evaluation Harness (18). Additionally, we evaluate the validation perplexity of the models following the same experimental settings described in FlashAttention (10; 8). We compare SLoPe against two state-of-the-art sparse pretraining methods, including (a) Wanda (51) $\rightarrow$ a one-shot pruning technique, (b) Extended SR-STE (61; 26) $\rightarrow$ a dynamic mask pretraining method for N:M sparsity, which serves as the foundation of follow-up work (27; 60; 26). Please note that SR-STE only supports stochastic gradient descent optimization, and FST extended it to other optimizers. We use the extension provided by FST in our work, and call it Extended SR-STE. The difference between Extended SR-STE and FST is that FST requires dense pretraining (fine-tuning) in the last 17% of pretraining and only prunes the MLP layers of the model, while SR-STE is fully sparse and prunes both the MLP and the Self-Attention layers of the model.

Figure 2 compare the validation perplexity and zero-shot accuracy of GPT2-Small and GPT2-Large across a range of sparse pretraining methods with different hyperparameters. We have additionally added lazy low-rank adapters to Extended SR-STE (61) to show the effectiveness of our approach in other methods and also compare both methods with more similar settings. While a gap in perplexity consistently exists between sparse and dense models, SLoPe achieves a lower perplexity compared to Wanda (51) and Extended SR-STE. Additionally, Table 3 summarizes the achieved accuracy of the models on zero-shot tasks, showing that SLoPe is consistently achieving a higher accuracy in comparison to Extended SR-STE. Moreover, adding lazy low-rank adapters can benefit both static and dynamic training methods. This improved accuracy stems from SLoPe's efficient allocation of the training budget. Specifically, Extended SR-STE, with its dynamic pruning masks, expends a significant portion of its training budget (e.g. gradient updates) updating weights that may be

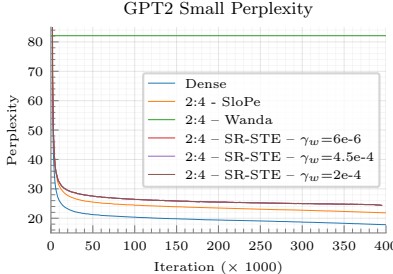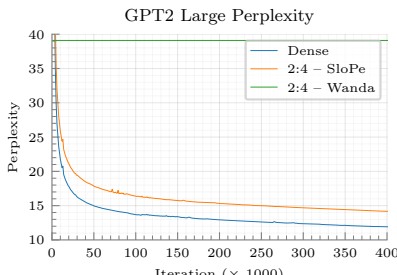

Figure 2: Validation perplexity of GPT2-Small and GPT2-Large on OpenWebText. $\gamma_w$ shows the value of the decay factor parameter in Extended SR-STE (FST).

Table 3: GPT2-Small accuracy results on zero-shot tasks. Adapter rank is the ratio of the low-rank adapter to the hidden dimension of the model. For Extended SR-STE, we have used a decay factor of 6e-6, since it resulted in the lowest perplexity in OpenWebText. The best performing sparse configuration is highlighted in bold.

| METHOD | ADAPTER RANK | MMLU ↑ | ARC CHALLENGE↑ | OPEN-BOOKQA↑ | WINO-GRANDE↑ | HELLA-SWAG↑ | MATHQA↑ | PIQA↑ | RACE↑ |
|---|---|---|---|---|---|---|---|---|---|
| DENSE | N/A | 22.9 | 20.7 | 16.2 | 50.6 | 28.5 | 21.8 | 59.8 | 28.4 |
| SLOPE | 2.1% | 23.0 | 19.3 | **16.4** | **50.8** | **27.5** | 20.8 | **57.6** | **27.2** |
|  | 0.05% | 23.0 | **19.4** | 16.2 | 50.5 | 27.4 | 20.8 | 57.5 | 27.1 |
|  | 0 | 23.0 | 19.3 | 16.0 | 50.1 | 27.5 | 20.8 | 57.4 | 27.1 |
| EXTENDED SR-STE | 2.1% | **24.2** | 18.3 | 14.2 | 47.5 | 26.9 | **21.4** | 55.2 | 24.2 |
|  | 0.05% | 24.1 | 18.4 | 14.2 | 47.5 | 26.8 | 21.2 | 54.5 | 24.2 |
|  | 0 | 24.1 | 18.3 | 12.6 | 47.5 | 26.9 | 21.2 | 54.8 | 24.0 |

ultimately pruned and not used at inference, leading to wasted resources. Appendix A provides further details and supporting evidence for this observation. Additional validation results for GPT experiments on GLUE dataset are also provided in Appendix Q and P.

**BERT-Large-Uncased.** We pretrain BERT-Large-Uncased (13) (355 M parameters) and fine-tune it for various question-answering and text classification tasks, following a similar approach to (42; 36; 44) for both pretraining and fine-tuning. Appendix G provides details on the pretraining and fine-tuning process. We evaluate the performance of BERT-Large-Uncased on the SQuAD v1.1 (48) and GLUE (57) tasks. We report the average metric score for GLUE and present the task-specific metrics in Appendix L. Please note that in all the experiments corresponding to BERT-Large-Uncased, when using Wanda, we have fine-tuned the model after pruning to improve the accuracy of the models, since using Wanda alone led to extremely low accuracy results.

**Effects of low-rank adapters.** To understand the impact of low-rank adapters on pretraining performance, we conducted ablations using low-rank adapter ranks of 4, 16, and 64 for 1% of the total number of iterations. These ranks represent up to 6.25% of the model's hidden dimension. Table 4 shows the results of these settings on SQuAD and GLUE downstream tasks. We present per-task metrics for GLUE in Appendix L. As expected, adding low-rank adapters improve the model's final accuracy across all tasks. Additionally, higher ranks improve the model's performance at the cost of increased computational requirements. It is also worth to note that incorporating low-rank adapters only in the final iterations (1% of total iterations) is sufficient to recover pretraining accuracy.

**Convergence rate of low-rank adapters.** We hypothesized that low-rank adapters would converge faster due to their significantly fewer learnable parameters. To test this, we introduced low-rank adapters in the second phase of BERT-Large-Uncased pretraining and monitored their convergence rate. Figure 3 shows the cosine similarity of the adapters, with the downsample adapter converging rapidly within 100 iterations and the upsample adapter converging slightly slower. Despite this, limiting training to 100 iterations still yields comparable results on downstream tasks.

**Effects of mixed N:M sparsity.** To study the sensitivity of different blocks to varying sparsity ratios and to assess their relative importance, we experiment across a range of configurations: (a) [2:4-2:4] → uniformly applying 2:4 sparsity across all layers (b) [2:4-2:8] → applying 2:4 sparsity

Table 4: SQuADv1.1 and GLUE results on BERT-Large-Uncased with different adapter ranks. $r$ denotes the ratio of the low-rank adapter to the hidden dimension (1024).

| DATASET | DENSE | $r = 0$ | $r = 0.39\%$ | $r = 1.56\%$ | $r = 6.25\%$ |
|---------|-------|---------|--------------|--------------|--------------|
| SQUAD | 90.44 | 89.1 | 89.1 | 89.2 | 89.5 |
| GLUE | 80.22 | 77.4 | 77.7 | 77.8 | 78.2 |

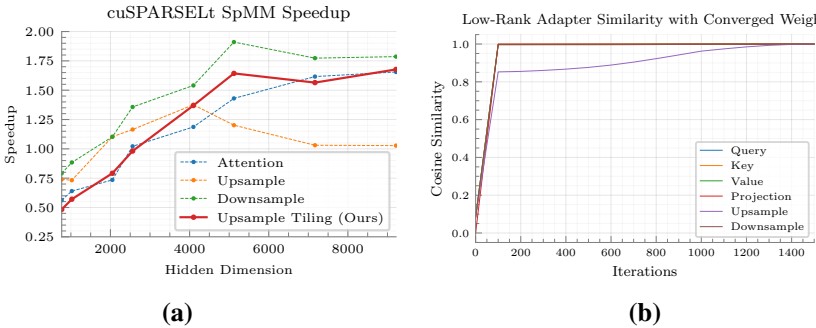

**(a)**      **(b)**

Figure 3: **(a)** The speedup achieved using cuSPARSELt backend in PyTorch for Attention ($d_{out} = d_{in}$), Upsample ($d_{out} = 4d_{in}$) and Downsample ($d_{out} = \frac{d_{in}}{4}$) matrices with a batch size of 2048. **(b)** The cosine similarity of the low-rank adapters and the converged adapters for different layers in the model. The cosine similarities are averaged among the 24 layers of BERT-Large-Uncased.

pattern to the first 12 blocks and a 2:8 sparsity pattern to the last 12 blocks and (c) [2:8-2:4] $\rightarrow$ we reverse the sparsity ratios for the first and last 12 blocks. Note that, to reduce computational costs, we use the same dense checkpoint for Phase-1 in all settings and a low-rank adapter of rank 40 for all models. We also replicate this experiment using Wanda (51) and report the comparison results.

Table 5: SQuADv1.1 results on BERT-Large-Uncased for different sparsity settings.

| SPARSITY PATTERN (FIRST 12 BLOCKS - LAST 12 BLOCKS) | SQUAD SLOPE | SQUAD WANDA | GLUE SLOPE | GLUE WANDA |
|---------|-------|-------|-------|-------|
| 2:4-2:4 | **90.17** | 89.93 | **79.08** | 78.84 |
| 2:4-2:8 | **89.85** | 89.55 | **79.03** | 77.24 |
| 2:8-2:4 | **89.67** | 86.57 | **75.92** | 69.08 |

Table 5 summarizes the GLUE and SQuAD results for these settings. As the results show, increasing the sparsity ratio reduces the accuracy of the model on all tasks. But when the first 12 blocks of the model are pruned, the accuracy drop is significantly higher, especially on the GLUE dataset. We conclude that the first blocks of the model are more sensitive to sparsity during pretraining, but one can sparsify the last blocks of LLMs more aggressively. We observe a similar pattern in Wanda results as well, but Wanda performs consistently worse than SLOPE in these cases.

**Effects of sparsification on different modules.** Each block in LLMs consists of a self-attention module and an MLP module, each containing multiple linear layers. We have analyzed the sensitivity of SLOPE to pruning each of those modules. Our results in Appendix F demonstrate that SLOPE can sustain competitive quality results while pruning all modules in the model.

## 4 CONCLUSION

In conclusion, SLOPE improves both pretraining and inference times while reducing memory footprint with negligible impact on model performance. SLOPE achieves these benefits by effectively using N:M sparsity and lazy low-rank adapters in both forward and backward passes, supported by an efficient design of CUDA kernels. Additionally, the use of lazy low-rank adapters allows for balancing memory footprint and model accuracy across a wide range models. The results show that SLOPE achieve up to 1.25$\times$ and 1.54$\times$ speedup for pretraining and inference, respectively. These speedups achieved while our method reduces the effective memory footprint by up to 0.63$\times$ (pretraining) and 0.61$\times$ (inference).

ACKNOWLEDGMENTS

This work was also supported in part by NSERC Discovery Grants (RGPIN-06516, DGECR00303), the Canada Research Chairs program, Ontario Early Researcher award, the Canada Research Chairs program, the Ontario Early Researcher Award, and the Digital Research Alliance of Canada (`www.alliancecan.ca`). Work of Zhao Zhang was supported by National Science Foundation OAC-2401246. We also acknowledge the Texas Advanced Computing Center (TACC) at The University of Texas at Austin for providing HPC resources that have contributed to the research results reported within this paper (`http://www.tacc.utexas.edu`). We extend our gratitude towards David Fleet, Karolina Dziugaite, Suvinay Subramanian, Cliff Young, and David Anugraha for reviewing the paper and providing insightful feedback. We also thank the extended team at Google DeepMind who enabled and supported this research direction.

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

# Appendix

## Table of Contents

## A    COMPARISON WITH DYNAMIC SPARSITY: SR-STE

We pretrained GPT2-Small (Section 3.2) using the SR-STE method (61) and reported the perplexity results in Figure 2. SR-STE aims to mitigate the Sparse Architecture Divergence (SAD) by dynamically adjusting the sparsity mask throughout training. We have tested various decay factor hyperparameters to find the optimal optimization strategy for SR-STE.

To understand the performance gap between SR-STE and SLoPe (our method) for the same training budget, we analyzed the mask dynamics in SR-STE. We plotted the *average number of mask elements*

changes during training compared to the final converged mask sparsity pattern. High mask change values indicate that training resources are spent on updating weights that ultimately get pruned and do not necessarily contribute to the final model accuracy.

Figure 4 shows this average mask difference per iteration relative to the converged model. As training progresses, the mask difference decreases, demonstrating SR-STE's convergence to a specific sparsity pattern. However, in SLoPe, where all resources are dedicated to optimizing weights under a *static* mask[4], SR-STE's dynamic approach leads to wasted computation (represented by the area under the curve in Figure 4). Consequently, for the same training budget, SLoPe achieves a lower perplexity in comparison to SR-STE due to its static mask approach.

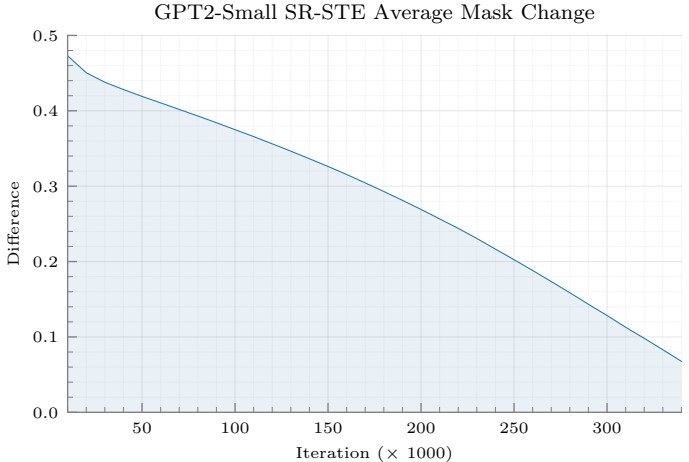

Figure 4: Average mask difference between each iteration and the converged sparsity pattern in GPT2-Small pretraining using SR-STE. The highlighted area shows the ratio of the resources used for updating weights that are pruned and not used in the inference of the model.

# B  cuSPARSELt Initialization Overhead: Static vs. Dynamic Sparsity

This section analyzes the time breakdown of the cuSPARSELt SpMM pipeline, highlighting the significant overheads associated with dynamically changing sparsity masks. The cuSPARSELt SpMM operation consists of two main phases: *(1) Setup* and *(2) Matrix Multiplication*. The setup phase involves initializing matrix handles and compressing the 2:4 sparse matrix. This compression copies non-zero values into a contiguous memory layout and generates indices for those values. The matrix multiplication phase leverages this metadata to perform the sparse matrix-matrix multiplication.

Figure 5 shows the setup and multiplication time for square matrices using the cuSPARSELt SpMM backend. As evident from the figure, the setup overhead is significantly larger than the actual matrix multiplication time. For SLoPe, which employs static sparsity masks, the setup cost is incurred only once and becomes negligible compared to the numerous matrix multiplications performed during training and inference. However, for dynamic sparsity patterns, such as Sparse-Dense Pretraining (26), Bidirectional Masks (60), and other similar methods(27; 52; 34; 61), this setup overhead can be substantial, leading to reduced speedup (as observed in Section 3.1 for Sparse-Dense Pretraining) or slowdowns in some configurations (as discussed in Appendix H).[5]

---

[4]We determine the pruning mask at the very first iteration and maintain it for the rest of training.
[5]A recent work observed a similar overhead using dynamic sparsity in cuSPARSELt SpMM pipeline (5).

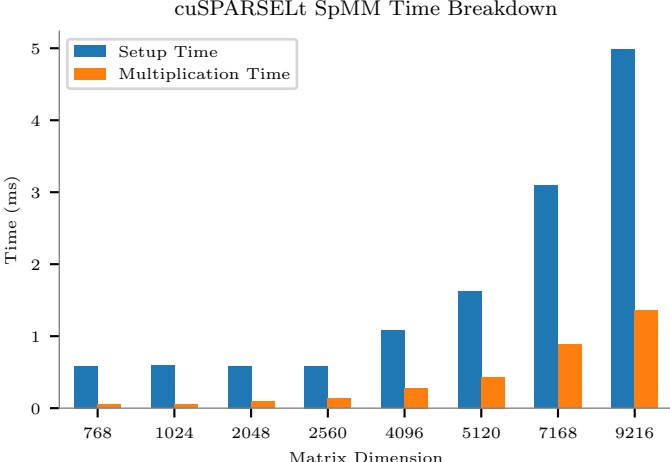

Figure 5: The setup and multiplication time for square matrices using the cuSPARSELt SpMM backend.

## C   LOW-RANK ADAPTER PERFORMANCE: SCALING AND ARITHMETIC INTENSITY

As discussed in Section 2.4, the computation time of low-rank adapters does *not* scale linearly with their rank. This section provides experimental results to illustrate this behavior in more detail. The computational complexity of low-rank matrix multiplications is $\mathcal{O}(brd)$, where $b$, $r$, and $d$ represent the batch size, low-rank, and input/output dimensions of the layer, respectively. Based on this complexity, we expect the computation time to be a linear function of $r$. In other words, reducing $r$ by a factor of $\alpha$ should result in a corresponding $\alpha$-fold reduction in computation time. However, in practice, this linearity does not hold. This deviation arises because the assumption underlying this expectation – that matrix multiplication is compute-bound – is not always true. Specifically, the arithmetic intensity of the operation can fall below the machine's balance point, as described in the Roofline model (59). Figure 6 shows the speedup achieved for different low-rank values using PyTorch's matrix multiplication function, which relies on the CUBLAS backend (39). The figure demonstrates that the achieved speedups are significantly lower than the ideal linear scaling, particularly when reducing the rank. Moreover, it is evident that as the matrix dimensions increase, the gap between the ideal speedup and the observed speedup diminishes. This behavior can be attributed to the increased arithmetic intensity for larger matrices, leading to better utilization of tensor cores.

## D   EFFICIENT LOW-RANK ADAPTER IMPLEMENTATION

As discussed in Section 2.4, a naïve implementation of low-rank adapters can lead to significant performance overheads due to the increased number of kernel launches and the low arithmetic intensity of their multiplications. To address these issues, we introduced two key optimizations: (1) concatenating one of the low-rank adapters with the sparse weights, and (2) fusing the multiplication of the other low-rank adapter with the subsequent result addition. These optimizations reduce kernel calls and increase arithmetic intensity, leading to more efficient utilization of GPU resources. Table 6 summarizes the speedup improvements achieved with these optimizations, demonstrating an inference speedup increase of up to 6%.

## E   EFFICIENT WEIGHT TILING IMPLEMENTATION

We observed that the dimensions and aspect ratios of matrices significantly influence system speedup (Section 2.4). To mitigate this, we implemented a matrix tiling strategy, dividing upsample matrices

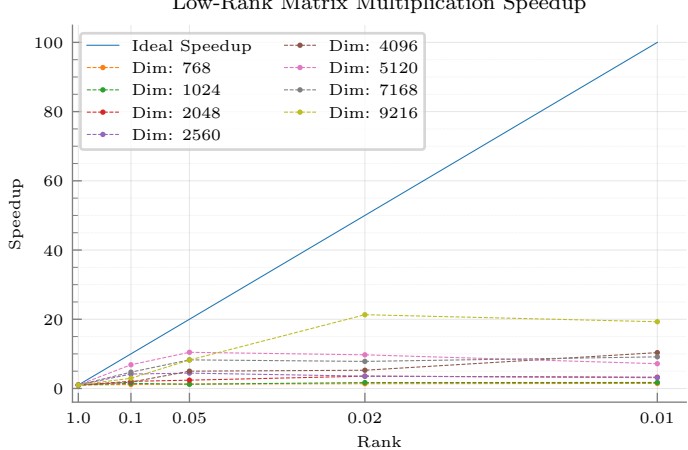

Figure 6: The speedup achieved by low-rank adapters in comparison to a dense matrix-multiplication.

Table 6: End-to-end speedup ($\times$) before (**left**) and after (**right**) efficient implementation of low-rank adapters.

| MODEL | INFERENCE 1.56% ADAPTER | INFERENCE 6.25% ADAPTER |
|---|---|---|
| OPT-66B | 1.15-1.20 | 1.12-1.19 |
| OPT-30B | 1.13-1.18 | 1.10-1.16 |
| OPT-13B | 1.11-1.10 | 1.09-1.10 |
| OPT-6.6B | 1.07-1.12 | 1.06-1.11 |
| OPT-2.6B | 1.01-1.06 | 0.97-1.00 |

into multiple square matrices. This approach significantly improves performance, as shown in Table 7. Our results demonstrate that matrix tiling can enhance training speed by up to 4% and inference speed by up to 12%, highlighting its effectiveness in optimizing system performance.

## F    SLoPe SENSITIVITY TO PRUNING DIFFERENT MODULE IN TRANSFORMER

LLMs typically consist of two main modules: the MLP and the self-attention. The attention module's weights are represented as a matrix in $\mathbb{R}^{d \times 3d}$, while the MLP uses weights in $\mathbb{R}^{d \times 4d}$ and $\mathbb{R}^{4d \times d}$, where $d$ denotes the hidden dimension. To investigate the impact of sparsity on these modules, we conducted two experiments during Phase-2 of BERT-Large-Uncased pretraining: (a) [MLP] $\rightarrow$ pruning both MLP and self-attention modules. Table 8 presents the SQuAD and GLUE results for these settings. As expected, we observe a consistent, albeit slight, decrease in model quality as more modules are sparsified. The marginal decrease in performance suggests that models are relatively insensitive to the specific modules being pruned when using our SLoPe pretraining method. This observation underscores the robustness of our approach and its ability to maintain competitive quality across diverse sparsity configurations.

## G    BERT-LARGE-UNCASED: PRETRAINING AND DOWNSTREAM EVALUATION

BERT-Large-Uncased pretraining consists of two phases, as illustrated in Figure 7. Phase 1 comprises 7,038 iterations with a global batch size of 65,536 and a sequence length of 128. Phase 2 includes 1,563 iterations with a global batch size of 32,768 and a sequence length of 512.

Table 7: End-to-end speedup ($\times$) before (**left**) and after (**right**) splitting the upsample matrix. In both cases, the optimization discussed in 6 is used.

| MODEL | TRAINING | INFERENCE NO ADAPTER | INFERENCE 1.56% ADAPTER | INFERENCE 6.25% ADAPTER |
|---|---|---|---|---|
| OPT-66B | 1.10-1.13 | 1.22-1.34 | 1.20-1.31 | 1.19-1.30 |
| OPT-30B | 1.09-1.14 | 1.23-1.32 | 1.18-1.28 | 1.16-1.27 |
| OPT-13B | 1.10-1.12 | 1.23-1.30 | 1.10-1.30 | 1.10-1.12 |
| OPT-6.6B | 1.08-1.08 | 1.21-1.19 | 1.12-1.13 | 1.11-1.12 |
| OPT-2.6B | 1.03-1.02 | 1.02-1.07 | 1.06-1.05 | 1.00-1.00 |

Table 8: SQuADv1.1 results on BERT-Large-Uncased for different pruned modules.

| PRUNED MODULES | SQUAD | GLUE |
|---|---|---|
| DENSE | 90.44 | 80.22 |
| MLP | 90.28 | 79.03 |
| MLP + SELF-ATTENTION | 89.35 | 77.72 |

Figure 7 shows the training loss for both phases under different sparsity settings. We observe that higher sparsity ratios generally lead to higher training loss in both phases. Interestingly, the loss/perplexity gap does not directly correlate with the observed accuracy drops in downstream tasks (4; 32; 50).

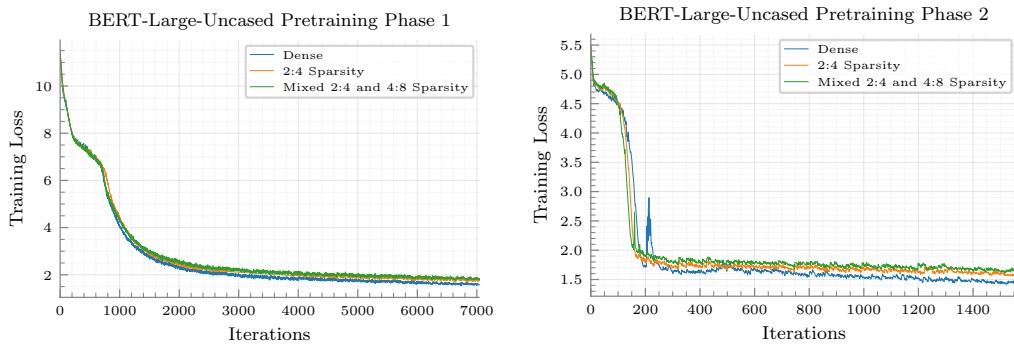

Figure 7: Training loss of BERT-Large-Uncased on WikiCorpus dataset for phase 1 and 2.

We evaluated the pretrained BERT-Large-Uncased models on the SQuAD v1.1 (48) and GLUE (57) benchmarks. SQuAD v1.1, a comprehensive question-answering dataset based on Wikipedia, is widely used for LLM training. We report the F1 score for SQuAD throughout the paper. GLUE, a diverse benchmark for natural language understanding tasks, provides a single aggregated score across various challenges, facilitating model comparisons. The paper presents the average metric score for GLUE, while task-specific metrics are detailed in Appendix L.

## H    PERFORMANCE OVERHEAD OF BIDIRECTIONAL MASK

Table 9 presents the runtime results of Bidirectional Masks (60), a state-of-the-art N:M sparsity method. Our analysis demonstrates that the mask search and associated overheads of this approach result in significant slowdowns compared to dense baselines. For these experiments, we utilized the repository provided in (60) and employed the same models used in their evaluation.

Table 9: End-to-end slow-down of Bi-directional Mask (60) in comparison to the dense baseline.

| MODEL | DATASET | SLOW-DOWN ($\times$) |
|---|---|---|
| MOBILENET V2 | CIFAR10 | 5.08 |
| RESNET-32 | CIFAR10 | 5.07 |
| VGG19 | CIFAR10 | 8.41 |
| RENET-18 | IMAGENET | 3.66 |
| RESNET-50 | IMAGENET | 3.01 |

## I SPARSITY RATIO ANALYSIS OF DOUBLE-PRUNED BACKWARD PASS

As described in Section 2.1, our proposed sparse pretraining approach involves pruning weights in both the forward and backward passes. During the backward pass, we apply both row-wise and column-wise pruning, which introduces additional zero values to the column-wise pruned weight matrices used in the forward pass. Theorem 2.1 demonstrates that the resulting sparsity ratio can be calculated using Equation 8. Figure 8 visualizes the imposed sparsity ratios for various N:M sparsity patterns. As expected, smaller N/M ratios lead to lower imposed sparsity ratios. Moreover, in most cases, the imposed sparsity ratio is significantly smaller than the original matrix's density ratio.

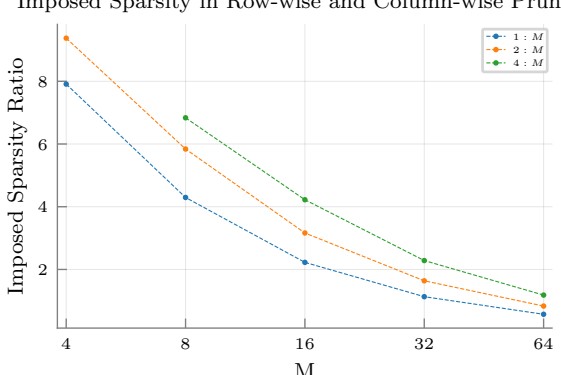

Figure 8: The imposed sparsity ratio when pruning the weight matrices in the backward pass.

## J SENSITIVITY TO THE CHOICE OF PRUNING MATRIX

In linear layers, three matrices are involved in the forward and backward passes: the input, the output gradient, and the weights. Pruning each of these matrices can have distinct effects on model performance.

To identify the optimal pruning strategy, we conducted an experiment where we pretrained GPT2-Small for 100,000 iterations (a quarter of the full pretraining) while systematically applying both static and dynamic pruning to each of the three matrices. Static pruning involves generating a random mask at initialization and applying it throughout training. Dynamic pruning, on the other hand, prunes matrices based on their magnitude at each iteration. For dynamic pruning, the dense matrix values are computed and stored, and then pruned at every step.

Figure 9 presents the validation perplexity for these experiments. Notably, pruning the output gradient led to model divergence after a few iterations and is not shown in the figure.

**Analysis.** As shown in Figure 9, static pruning consistently achieved lower perplexities. This behavior suggests that focusing computational resources on elements that remain active throughout training

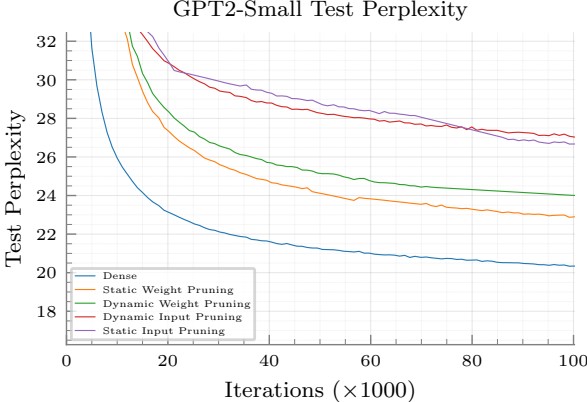

Figure 9: Validation perplexity on GPT2-Small pretraining for 100,000 iterations for different matrix pruning settings. Pruning the output gradients leads to divergence within the few iterations and hence is not reported.

can lead to improved performance. Furthermore, pruning weights resulted in lower perplexities compared to pruning inputs, indicating that weights are generally a better target for pruning.

**Intuition.** Pruning weights is analogous to removing connections between neurons. Pruning activation tensors is similar to introducing a non-linear function (akin to max-pooling) before each linear layer. Pruning output gradients, however, lacks practical justification and introduces errors into the backward pass, leading to model divergence.

## K    IMPLEMENTATION DETAILS

This section details the implementation of the custom functions and CUDA kernels used in Algorithm 1 to facilitate efficient sparse training.

**Initialization, sparse matrix setup, and SpMM kernels.** Before utilizing the cuSPARSELt APIs, a crucial initialization phase ensures proper configuration of essential variables for our computational task. Following initialization, we configure the sparse data formats tailored for sparse matrices. This involves initializing matrix descriptors, pruning the matrices, and compressing them into a more compact representation. cuSPARSELt employs an automated search to determine the optimal kernel for executing SpMM. While setting up these sparse data formats incurs a non-negligible computational cost, this overhead is mitigated by the repetitive nature of matrix multiplications during the training process.

**Prune and compress.** The gradient of the loss function with respect to the weights requires pruning using the same mask as the weight matrix. Consequently, it contains 50% extra zero values in the dense format. To address this redundancy, we developed an optimized CUDA kernel, integrated into PyTorch, that masks the gradients accordingly, eliminating the storage of unnecessary data and reducing memory usage. The output of this operation is a new matrix in $\mathbb{R}^{d_{out} \times \frac{d_{in}}{2}}$.

**Sparse matrix addition.** The cuSPARSELt sparse data format does not natively support addition operations. However, for matrices $A$ and $B$ sharing the same sparsity patterns, we developed an optimized CUDA kernel seamlessly integrated into the PyTorch training workflow. This kernel efficiently computes linear combinations of the form $\beta A + \gamma B$, where $\beta$ and $\gamma$ are arbitrary user-defined constants. This functionality is particularly useful for adding sparse weights to gradients in optimizers that utilize weight decay.

**Update Sparse Matrix.** After the optimizer updates the weight tensor values based on its rules, we need to update the sparse matrix format to reflect these changes. We implemented an optimized

Table 10: GLUE results for each task in the experiments discussed in section 3.

| Method | Phase | Rank | First 12 Blocks | Last 12 Blocks | CoLA (mcc) | SST-2 (acc) | MRPC (f1) | STS-B (corr) | QQP (f1) | RTE (acc) | MNLI (acc) | QNLI (acc) |
|--------|-------|------|-----------------|----------------|------------|-------------|-----------|--------------|----------|-----------|------------|------------|
| Dense | 1,2 | 0 | 2:4 | 2:4 | 51.6 | 91.9 | 81.2 | 87.5 | 87.8 | 66.4 | 84.1 | 91.3 |
| SLoPe MLP Mixer Only | 2 | 0 | 2:4 | 2:4 | 41.8 | 91.4 | 88.7 | 87.2 | 85.9 | 65 | 82.1 | 90.1 |
| SLoPe MLP Mixer + Self-Attention | 2 | 0 | 2:4 | 2:4 | 38.8 | 90.4 | 85.9 | 86.4 | 85.9 | 63.5 | 81.5 | 89.3 |
| SLoPe with Non-Lazy Adapters | 2 | 40 | 2:4 | 2:4 | 43.3 | 90.8 | 89 | 87 | 86 | 64.6 | 82.3 | 89.6 |
| SLoPe with Non-Lazy Adapters | 2 | 40 | 2:8 | 2:4 | 29 | 89.7 | 83.7 | 85.6 | 85.2 | 66.8 | 79.9 | 87.4 |
| SLoPe with Non-Lazy Adapters | 2 | 40 | 2:4 | 2:8 | 44.1 | 91.1 | 89.8 | 86.6 | 86.3 | 62.5 | 82.3 | 89.6 |
| SLoPe | 1,2 | 0 | 2:4 | 2:4 | 37.9 | 91.4 | 85.4 | 86.6 | 85.8 | 62.5 | 80.7 | 88.6 |
| SLoPe | 1,2 | 4 | 2:4 | 2:4 | 38.5 | 91.4 | 85.8 | 86.8 | 85.8 | 63.9 | 80.8 | 88.4 |
| SLoPe | 1,2 | 16 | 2:4 | 2:4 | 39.2 | 91.3 | 86.4 | 86.6 | 86 | 63.5 | 80.8 | 88.2 |
| SLoPe | 1,2 | 64 | 2:4 | 2:4 | 42.7 | 90.3 | 85.1 | 86.8 | 85.7 | 66.4 | 80.3 | 88.5 |
| WANDA | N/A | 0 | 2:4 | 2:4 | 43.0 | 91.4 | 88.3 | 86.9 | 86.1 | 63.5 | 81.9 | 89.6 |
| WANDA | N/A | 0 | 2:8 | 2:4 | 4.6 | 0.88 | 81.3 | 81 | 83.3 | 53.8 | 76.7 | 83.9 |
| WANDA | N/A | 0 | 2:4 | 2:8 | 42.1 | 91.7 | 84.4 | 87.2 | 85.6 | 63.5 | 81.5 | 81.9 |

CUDA kernel that copies the weight tensors from the PyTorch format into the cuSPARSELt data type, enabling efficient storage and manipulation of sparse weights.

## L  TASK-SPECIFIC GLUE RESULTS

The GLUE benchmark (57) comprises eight distinct natural language understanding classification tasks. While Section 3 presented the average GLUE score as a measure of overall model performance, this section provides a more detailed analysis by presenting the complete task-specific results for each training setting in Table 10.

## M  INTEGRATION WITH FLASH ATTENTION

To show the compatibility of SLoPe with other optimization methods, we test integrate SLoPe with FlashAttention-2 (8) and show that these approaches are orthogonal in practice and can boost the performance of the model separately. Table 11 summarizes the speedup achieved with and without SLoPe or FlashAttention-2. As it can be observed, each of these methods can improve the speed of the model both in training and inference, and adding them together will increase the speedup even further.

## N  ADDITIONAL RELATED WORK

**Model pruning.** Pruning the models has been one of the most effective methods to reduce the complexity of LLMs (24). One can pretrain the LLMs sparsely (14) or the pruning can happen after a dense pretraining (22; 30), possibly followed by a fine-tuning stage to recover part of the lost accuracy (16; 20). Pruning the models after pretraining can be costly (49; 21) and typically fails to maintain their accuracy (15; 51). While the sparse pretraining methods improve the accuracy of the model, they either use unstructured sparsity patterns that cannot be accelerated with the current hardware

Table 11: Speedup of SLoPe and FlashAttention-2 (FA2) on OPT models.

| MODEL SIZE | TRAINING | | | INFERENCE | | | INFERENCE | | INFERENCE | |
|---|---|---|---|---|---|---|---|---|---|---|
| | FA2 | SLoPe | SLoPe + FA2 | FA | SLoPe | SLoPe + FA2 | FA2 | SLoPe + FA2 | FA2 | SLoPe + FA2 |
| 66B | 1.28 | 1.13 | 1.53 | 1.36 | 1.34 | 1.99 | 1.31 | 1.95 | 1.30 | 1.91 |
| 30B | 1.36 | 1.14 | 1.66 | 1.46 | 1.32 | 2.24 | 1.28 | 2.24 | 1.27 | 2.20 |
| 13B | 1.47 | 1.12 | 1.84 | 1.61 | 1.30 | 2.48 | 1.30 | 2.24 | 1.12 | 2.19 |
| 6.7B | 1.60 | 1.08 | 1.94 | 1.71 | 1.21 | 2.50 | 1.13 | 2.50 | 1.12 | 2.45 |
| 2.6B | 2.26 | 1.05 | 2.56 | 2.47 | 1.07 | 3.23 | 1.05 | 3.09 | 1.00 | 2.92 |

(55) or have significant overheads when searching for and applying their structured sparse masks (27; 60; 52).

**Low-rank adapters.** Low-rank adapters have emerged as a promising method to reduce the fine-tuning costs associated with pre-trained LLMs and enable more efficient task switching (25). Different quantization and initialization schemes have been proposed to reduce their overheads in LLM fine-tuning (12; 19). Adding low-rank factors to sparse matrices is a low-weight mechanism widely used to improve the accuracy of approximations of dense matrices (2). In machine learning, the sparse plus low-rank approximations are limited to attention heads (37; 3) and pruning after pretraining (38; 31), and the sparse plus low-rank pretraining has not been investigated. Additionally, the sparse plus low-rank fine-tuning work does not provide acceleration in both forward and backward pass of the fine-tuning process. Furthermore, the low-rank adapters in these works are added at the beginning of the fine-tuning process, adding extra overheads to the fine-tuning process.

## O   EXPERIMENT SETUP, HYPERPARAMETERS, COMPUTE RESOURCES

Our experiments were conducted on the Narval and Mist clusters at Compute Canada (7) and the Lonestar 6 cluster at the Texas Advanced Computing Center (54). Each Narval node is equipped with four Nvidia A100 GPUs, each with 40GB of memory. Mist nodes feature four Nvidia V100 GPUs, each with 32GB of memory, while Lonestar 6 nodes have three Nvidia A100 GPUs, each with 40GB of memory. For our accuracy experiments, we emulated 2:4 and N:M sparsity using custom-designed, low-overhead CUDA kernels to prune weights in both the forward and backward passes. We utilized a mixture of available resources across the clusters, as model accuracy is not hardware-dependent.

Our speedup and memory saving experiments were conducted on a single A100 GPU in the Narval cluster. We ran 1000 iterations of training or inference to gather the necessary statistics. For speedup experiments, we reported the median of the 1000 samples to mitigate the effects of outliers. Each memory reduction experiment was run five times, and the median value was reported. We employed the default hyperparameters found in the NVIDIA BERT codebase (42) and the FlashAttention GPT codebase (10; 8). Further tuning of hyperparameters for sparse pretraining is left as a future direction. Training BERT-Large-Uncased required approximately 32 hours on 64 A100-64GB GPUs. The pretraining of GPT2-Small/Large took 32 and 111 hours, respectively, on 64 V100-32GB GPUs.

## P   COMPARISON WITH DENSE MODELS

To compare the performance of sparse models with dense models of the same size, we have conducted an experiment with GPT2-Small, in which we have reduced the number of transformer blocks in the model to half of GPT2-Small. We call this new configuration GPT2-Half. Tables 12 and 13 summarize the accuracy results for GPT2-Half on different zero-shot downstream tasks.

It can be observed that SLoPe outperforms GPT2-Half on average, while dynamic sparse training methods, such as SR-STE perform worse than it. Additionally, it is clear that adding low-rank adapters to the model improves the accuracy of all sparse pretraining methods.

Table 12: Performance comparison across different GPT models, sparsity methods, and LoRA ranks on various tasks. E-SR-STE stands for Extended SR-STE.

| Model | Method | LoRA (r) | MMLU | Arc Challenge | Open Book QA | Average |
|---|---|---|---|---|---|---|
| GPT2-Small | Dense | r = 0 | 22.9 | 20.7 | 16.2 | 19.94 |
| GPT2-Small | SLOPE | r = 0 | 23.0 | 19.3 | 16.0 | 19.43 |
| GPT2-Small | SLOPE | r = 0.05% | 23.0 | 19.4 | 16.2 | 19.53 |
| GPT2-Small | SLOPE | r = 2.1% | 23.0 | 19.3 | 16.4 | 19.57 |
| GPT2-Small | E-SR-STE | r = 0 | 24.1 | 18.3 | 12.6 | 18.33 |
| GPT2-Small | E-SR-STE | r = 0.05% | 24.1 | 18.4 | 14.2 | 18.90 |
| GPT2-Small | E-SR-STE | r = 2.1% | 24.2 | 18.3 | 14.2 | 18.90 |
| GPT2-Half | Dense | r = 0 | 22.9 | 19.5 | 16.0 | 19.47 |

## Q   ZERO-SHOT GLUE RESULTS FOR GPT

We have tested the accuracy of the models on the zero-shot GLUE tasks in Language Model Evaluation Harness (18). Table 13 summarizes the achieved GLUE results by different models. It can be seen that SLOPE outperforms other SR-STE and GPT2-Half on average. Additionally, SR-STE performs better than GPT2-Half in GLUE task.

Table 13: Performance comparison of GPT models using different sparsity methods and LoRA ranks on GLUE tasks. E-SR-STE stands for Extended SR-STE.

| Model | Method | LoRA (r) | CoLA | MNLI (m) | MNLI (mm) | MRPC | QNLI | QQP | RTE | SST2 | Avg |
|---|---|---|---|---|---|---|---|---|---|---|---|
| GPT2-Small | Dense | r = 0 | 0 | 32.4 | 33.2 | 66.9 | 50.3 | 51.8 | 49.8 | 59.3 | 43.2 |
| GPT2-Small | SLOPE | r = 0 | 0 | 34.3 | 34.0 | 72.5 | 50.0 | 48.5 | 50.0 | 52.3 | 42.8 |
| GPT2-Small | SLOPE | r = 0.05% | 0 | 34.3 | 34.1 | 72.6 | 49.8 | 48.8 | 50.9 | 52.3 | 42.9 |
| GPT2-Small | SLOPE | r = 2.1% | 0 | 34.3 | 34.0 | 71.6 | 50.0 | 49.0 | 52.0 | 52.6 | 43.1 |
| GPT2-Small | E-SR-STE | r = 0 | 0 | 33.6 | 33.9 | 57.1 | 50.7 | 50.4 | 55.2 | 54.7 | 42.5 |
| GPT2-Small | E-SR-STE | r = 0.05% | 0 | 33.1 | 33.6 | 57.9 | 51.0 | 50.5 | 55.4 | 55.0 | 42.6 |
| GPT2-Small | E-SR-STE | r = 2.1% | 0 | 33.3 | 33.5 | 58.2 | 51.0 | 50.5 | 55.2 | 55.2 | 42.6 |
| GPT2-Half | Dense | r = 0 | 0.0 | 33.9 | 33.8 | 53.6 | 51.1 | 47.7 | 56.7 | 50.6 | 41.1 |

## R   EXTENDED SR-STE AND FST  IMPLEMENTATION DETAILS

 Before we proceed with the details of Extended SR-STE and FST, we clarify the notations used in our manuscript and the FST  paper (26) in table 14

Table 14: Description of Key Terms

| Term | Description |
|---|---|
| Sparse Pretraining | Common notation used in SLoPe and FST, indicating the use of sparse weights during pretraining. |
| Dense Finetuning | Notation used in the FST paper, indicating an extended pretraining phase. |
| Downstream Finetuning | Performance after pretraining concludes, used to finetune the model for specific downstream tasks. |
| FST | Extended pretraining technique focused on dense finetuning. |
| Extended SRT | Variation of sparse pretraining extended with additional finetuning. |

 We compare SLOPE with FST  exclusively for *training speedups and memory savings* (Table-1-Page-7 and Table-2-Page-8). Comparing pretraining quality between SLOPE and FST  is less meaningful because the final models produced by these methods differ significantly in the number of parameters. Specifically, FST  produces a dense model after sparse pretraining and dense finetuning (99% sparse pretraining + 1% sparse + low-rank adaptation), while SLOPE produces a sparse model augmented

with lightweight low-rank adapters. The number of parameters in the FST model is approximately $2\times$ larger than in SLoPe, which makes a direct quality comparison imbalance.

We compare SLoPe with Extended SR-STE in terms of model quality, focusing on understanding the dynamics between static and dynamic masking under an equal number of parameters. This allows for a fair, "apple-to-apple" comparison between the methods (iso-params). We refer to this method as "Extended SR-STE" because, while the original SR-STE approach was designed for use with SGD, the FST paper extended it to support other optimizers.

The FSTand Extended SR-STE code are available in Listing 1 and Listing 2 respectively.

```python
def forward(ctx, x, weight, weight_sparse, weight_sparse_T, bias):
    ctx.save_for_backward(input, weight_sparse_T, bias)
    ctx.shape = x.shape
    x = x.view(-1, x.shape[-1])
    output = torch.mm(x, weight_sparse.t())
    if bias is None:
        return output.view(*ctx.shape[:-1], -1)
    else:
        return output.view(*ctx.shape[:-1], -1) + bias

def backward(ctx, grad_output):
    grad_output = grad_output.half()
    x, weight_T, bias = ctx.saved_tensors
    grad_input = grad_weight = grad_bias = None
    if ctx.needs_input_grad[0]:
        if grad_output.stride() == (0, 0, 0):
            grad_output = torch.ones_like(grad_output, device=grad_output
                .device, dtype=grad_output.dtype)
        grad_output = grad_output.view(-1, grad_output.shape[-1])
        grad_input = torch.mm(grad_output, weight_T.t()).view(
            ctx.shape)
    if ctx.needs_input_grad[1]:
        x = x.view(-1, input.shape[-1])
        grad_output = grad_output.view(-1, grad_output.shape[-1])
        grad_weight = torch.mm(to_sparse_semi_structured(grad_output.t(),
            MVUE24=True), x)
    if ctx.needs_input_grad[2]:
        grad_bias = grad_output.sum(0)
    return grad_input, grad_weight, None, None, grad_bias
```

Listing 1: FST Algorithm

```python
def forward(ctx, input, weight, mask, weight_factor):
    sparse_weight = weight.clone().detach()
    sparse_weight[mask] = 0.
    ctx.save_for_backward(input, sparse_weight, weight_factor * mask *
        weight)
    output = torch.matmul(input, sparse_weight.t())

    output = output.clone()
    return output

@staticmethod
def backward(ctx, grad_output):
    input, weight, weight_addition_term = ctx.saved_tensors
    input_shape = input.shape
    if input.dim() == 3:
        new_batch_size = input_shape[0] * input_shape[1]
        input = input.reshape(new_batch_size, -1)
        grad_output = grad_output.reshape(new_batch_size, -1)
    grad_output, grad_output_mask = prune_column_wise(grad_output)
    grad_weight = torch.matmul(grad_output.t(), input)
    grad_weight += weight_addition_term
```

```
22    grad_input = torch.matmul(grad_output, weight)
23    grad_input = grad_input.reshape(input_shape)
24    return grad_input, grad_weight, None
```

Listing 2: Extended SR-STE Algorithm. The weights are stored as dense and are pruned on-the-fly.

## S  COMPARISON OF DEPTH AND WIDTH PRUNING

Depth pruning refers to reducing the number of layers in a model, while width pruning means reducing the size of the weights inside each layer in the model. We have conducted an experiment with depth and width pruning on LLaMA-2-7B (56) and Gemma-2-2B and Gemma-2-9B (53) to compare the effects of depth and width pruning on the performance of the models. The configurations used for the models are summarized in Tables 15, 17, 16. Similar to (26), we reduced the aspect ratio of the Up-Sample and Down-Sample modules to half. Please note that this mechanism gives an advantage to width pruning methods, as the number of parameters in the Self-Attention modules remain intact. Our experiments show that

Table 15: Model Configurations for LLaMA-2 7B

| Pruning Method | Attributes |
|---|---|
| Baseline | base_emb_dim: 4096
base_num_query_heads: 32
base_num_kv_heads: 32
base_mlp_dim: 11008
base_num_decoder_layers: 32
head_dim: 128 |
| Depth Pruning | base_emb_dim: 4096
base_num_query_heads: 32
base_num_kv_heads: 32
base_mlp_dim: 11008
base_num_decoder_layers: 16 # half the number of layers
head_dim: 128 |
| Width Pruning | base_emb_dim: 4096
base_num_query_heads: 32
base_num_kv_heads: 32
base_mlp_dim: 5504 # half the number of dimensions
base_num_decoder_layers: 32
head_dim: 128 |

Preliminary retraining loss curves, as shown in Figure 10 suggest no significance difference between depth-pruning and width-pruning during pretraining. Interestingly, in some cases, depth-pruning appears to outperform width-pruning.

## T  PROOFS

### T.1  LEMMA 2.1

*Proof.* Considering a matrix with $N : M$ column-wise pruned sparsity pattern, we want to prune the matrix using $N : M$ sparsity pattern row-wise as well. Let's define random variable $X$ as the number of added non-zeros to $M$ row-wise consecutive elements and $Y$ as the number of non-zeros in $M$ row-wise consecutive elements.

$$E[X] = \sum_{i=1}^{M-N} Pr[X = i]i \tag{12}$$

Replacing $Pr[X = i] = Pr[Y = N + i]$ in Equation 12, we will get Equation 13, where we used a change in dummy variable $j = N + i$.

Table 16: Model Configurations for Gemma-9B

| Pruning Method | Attributes |
|---|---|
| Baseline | base_emb_dim: 3584
base_num_query_heads: 16
base_num_kv_heads: 8
base_mlp_dim: 14336
base_num_decoder_layers: 20 # merged local and global attention
head_dim: 256 |
| Depth Pruning | base_emb_dim: 3584
base_num_query_heads: 16
base_num_kv_heads: 8
base_mlp_dim: 14336
base_num_decoder_layers: 10 # half the merged layers
head_dim: 256 |
| Width Pruning | base_emb_dim: 3584
base_num_query_heads: 16
base_num_kv_heads: 8
base_mlp_dim: 7168 # half the number of dimensions
base_num_decoder_layers: 20 # merged local and global attention
head_dim: 256 |

Table 17: Model Configurations for Gemma-2B

| Pruning Method | Attributes |
|---|---|
| Baseline | base_emb_dim: 2304
base_num_query_heads: 8
base_num_kv_heads: 4
base_mlp_dim: 9216
base_num_decoder_layers: 12 # merged local and global attention
head_dim: 256 |
| Depth Pruning | base_emb_dim: 2304
base_num_query_heads: 8
base_num_kv_heads: 4
base_mlp_dim: 9216
base_num_decoder_layers: 6 # half the merged layers
head_dim: 256 |
| Width Pruning | base_emb_dim: 2304
base_num_query_heads: 8
base_num_kv_heads: 4
base_mlp_dim: 4608 # half the number of dimensions
base_num_decoder_layers: 12 # merged local and global attention
head_dim: 256 |

$$E[X] = \sum_{i=1}^{M-N} Pr[Y = N + i]i = \sum_{j=N+1}^{M} Pr[Y = j](j - N) \tag{13}$$

Considering the definition of $Y$, it can be inferred that random variable $Y$ has binomial distribution with a success probability of $\frac{N}{M}$. As a result Equation 14 shows the probability mass distribution of $Y$.

$$Pr[Y = j] = \binom{M}{j} s^j (1 - s)^{M-j}; s \triangleq \frac{N}{M} \tag{14}$$

By replacing Equation 14 in Equation 13, we will get Equation 15.

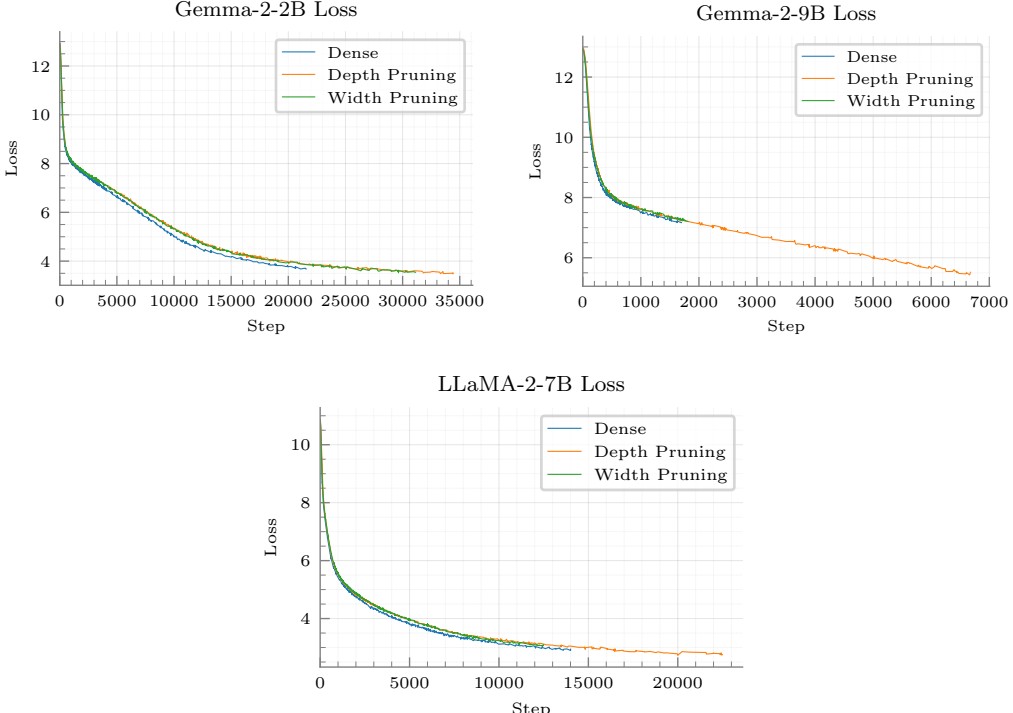

Figure 10: Comparison of the loss of depth and width pruning methods.

$$E[X] = \sum_{j=N+1}^{M} \binom{M}{j} ] s^j (1-s)^{M-j} (j-N) \tag{15}$$

Let's define random variable $Z$ as the added sparsity ratio to the matrix by the extra pruning. Since $X$ was the number of added non-zeros in $M$ consecutive elements, $E[Z] = \frac{1}{M} E[X]$, and hence Equation

$$E[Z] = D(A^R) - D(A^{R,C}) = \sum_{j=N+1}^{M} \binom{M}{j} s^j (1-s)^{M-j} \frac{j-N}{M} \tag{16}$$

## T.2   THEOREM 2.2

*Proof.* In an optimization problem, we are aiming to find the optimal solution to Equation 17.

$$\min_{W_i} E_X[\mathcal{L}(X, W_i)] \tag{17}$$

When using backpropagation, which is based on the chain rule in derivation, we compute the gradient in Equation 18.

$$.E_X[\nabla_{X_i} \mathcal{L}(X, W_i)] = E_X[\nabla_{Y_i} \mathcal{L} W] \tag{18}$$

Let's define random variable $M$ as a uniformly random mask of 0's and 1's. The mask will be 1 at each point with a probability of $\frac{N}{M}$. Let's define $O \triangleq E[M]$. $O$ is a matrix of all $\frac{N}{M}$'s. As a result $O \odot W = \frac{N}{M} A$ for an arbitrary matrix $A$.

$$E_X[\nabla_{Y_i}\mathcal{L}W] = E_X[\nabla_{Y_i}\mathcal{L}(\frac{M}{N}O \odot W)] = E_X[\nabla_{Y_i}\mathcal{L}(\frac{M}{N}E_M M \odot W)] \tag{19}$$

By using the linearity of derivation and expectation operators, we can get the result in Equation 20, which proves the theorem.

$$E_X[\nabla_{X_i}\mathcal{L}(X, W_i)] = \frac{M}{N}E_M[E_X[\nabla_{Y_i}\mathcal{L}(M \odot W)]] \tag{20}$$

