# OpenReview forum: "SLoPe: Double-Pruned Sparse Plus Lazy Low-Rank Adapter Pretraining of LLMs"
_ICLR.cc/2025/Conference — ICLR 2025 Poster_

### Official Review · Reviewer_B3ze · 2024-10-27

**Soundness:** 3
**Presentation:** 2
**Contribution:** 3
**Rating:** 6
**Confidence:** 4

**Summary:**

This paper proposes SLoPE, a Double-Pruned Sparse Plus Lazy Low-rank Adapter pretraining method that improves the accuracy and accelerates training and inference of sparse LLMs. By incorporating a low-rank adapter during the final 1% of pretraining iterations, the model accuracy is well maintained. Experimental results show that the proposed method significantly speedups in both pretraining and inference phases and efficiently reduces memory footprint.

**Strengths:**

1. The authors propose a double-pruned backward pass to improve model quality and reduce mask search overhead. They introduce additional low-rank adapters in the final 1% iterations of pretraining and improving model capacity.

2. The paper is well-written and easy to follow up. The finding of adding low-rank adapter in the last 1% iterations achieving better performance is interesting!

**Weaknesses:**

1. Why does introducing additional zeros in an N:M pattern during the computation of the input gradient in the backward pass of the first 99% of iterations not severely affect accuracy?

2. The memory footprint for the pretraining should be higher than for inference, so why are the memory footprint reduction multiples similar? In the backward pass, calculating weight gradient involves a dense computation. While double-pruning reduces the computation of input gradients, the overall memory reduction should be more substantial.

3. As discussed in section 2.2, the authors introduce a hyperparameter to balance the memory footprint, computational efficiency, and model quality. In the experiments on the effects of low-rank adapters, ranks are set to 4,16, and 64. Why weren’t larger ranks used? Since the low-rank adapter is only applied in the final 1% of iterations, the computational overhead should be minimal.

**Questions:**

Please see above.

---

> ### Author Response · Authors · 2024-11-17
> **Gradient Accuracy, Memory Footprint Analysis, and Low-Rank Adapter Hyperparameters**
>
> We thank you for your feedback and positive assessment of our work. Below, we address your specific questions in detail.
>
> # Impact of Introducing Additional Zeros in N:M Pattern
>
> Thank you for this question. As demonstrated in Theorem 2.2 (Page 4), by varying the backward pass mask across iterations, the *expected value of the gradients* matches the original unpruned gradient. This ensures that introducing additional zeros does not significantly degrade accuracy. To provide an intuitive explanation, in each backward pass, a subset of weights is pruned. By dynamically changing the location of these zeros across iterations, the SLoPe input gradients converged toward the original input gradients over time, maintaining accuracy.
>
> ---
>
> # Memory Footprint Reduction During Pretraining and Inference
>
> You raise an important point regarding memory footprint reduction. During pretraining, we store both the weights and their transposed copies to enable the double-pruned backward pass. This introduces additional memory overhead, narrowing the gap between the end-to-end memory reduction for pretraining (68\%) and inference (54\%).
>
> Below, we provide a detailed analysis of memory savings per four consecutive elements, which can be generalized to larger model sizes. This analysis has been included in Revised-Paper-Page-7.
>
> ## Inference Memory Reduction:
>
> ### Dense Model:
>
>     Weight: 4 * 16 bits
>
> ### Sparse Model:
>
>     Weight: 2 * 16 bits (Non-zeros) + 3 bits (Indices)
>
> ### Memory Saving (2:4 Sparse vs. Dense)
>
>     (2 * 16 + 3) / (4 * 16) = 54%
>
> ---
>
> ## Training Memory Reduction:
>
> ### Dense Model:
>
>     Weight: 4 * 16
>
>     Gradient: 4 * 16
>
>     Optimizer States: 2 * 4 * 32
>
> ### Sparse Model with 2:4 Pattern:
>
>     Weight: 2 * 16 bits (Non-zeros) + 3 bits (Indices)
>
>     Weight-Transpose: 2 * 16 bits (Non-zeros) + 3 bits (Indices)
>
>     Binary Mask: 4 * 8 bits
>
>     Gradient: 2 * 16 bits
>
>     Optimizer States: 2 * 2 * 32
>
> ### Memory Saving (2:4 Sparse vs. Dense)
>
>     (2 * 16 + 3 + 2 * 16 + 3 + 4 * 8 + 2 * 16 + 2 * 2 * 32) / (4 * 16 + 4 * 16 + 2 * 4 * 32) = 68%
>
> ---
>
> # Choice of Low-Rank Adapter Sizes
>
> Thank you for your observation regarding the size of low-rank adapters. To maximize the speedups offered by SLoPe during inference, it is essential to minimize the overhead introduced by low-rank adapters. This consideration led us to select relatively small low-rank adapter sizes compared to the model’s hidden dimensions. This choice ensures that the computational overhead remains minimal while maintaining performance benefits.
>
> ---
> We hope these responses address your concerns and provide additional clarity. Thank you again for your valuable feedback.

---

> > ### Comment · Reviewer_B3ze · 2024-11-26
> >
> > Thanks for your response. After reading other reviewers' responses, I will maintain my score.

---

> > > ### Author Response · Authors · 2024-11-28
> > > **Thank you! Additional clarifications**
> > >
> > > Dear Reviewer B3ze,
> > >
> > > Thank you for taking the time to review our work and for providing your feedback. We appreciate your effort in evaluating our submission.
> > >
> > > We noticed that you have decided to maintain your score after reading the rebuttal and other reviewers' responses. We would like to highlight that we have provided a detailed response to [Reviewer NKfR](https://openreview.net/forum?id=lqHv6dxBkj&noteId=1lfuUA3d4g) in three parts ([Part1](https://openreview.net/forum?id=lqHv6dxBkj&noteId=dAl1Bi4ccu), [Part2](https://openreview.net/forum?id=lqHv6dxBkj&noteId=IXrHn9aK3I), and [Part3](https://openreview.net/forum?id=lqHv6dxBkj&noteId=kOvkHigUk1)), addressing their concerns comprehensively and also revised the manuscript to:
> > >
> > > 1. Provide results for five additional zero-shot results (in total eight) (`Section-3.2-Table-3-Page-9`)
> > > 2. Compared depth-pruning vs. width-pruning (`Appendix-S-Figure-10-Page-29`)
> > > 3. Added clear notations for different techniques, SLoPe, Extended SR-STE, and FST (`Appendix-Table-14-Page-25`)
> > >
> > > We believe these clarifications should resolve all the points raised. If there are specific aspects of our responses or discussion that you find unconvincing, we would greatly appreciate your feedback. Identifying the areas where additional clarity is needed will allow us to address any lingering concerns effectively.
> > >
> > > With the discussion period now **extended to December 2nd**, we are hopeful that we can continue this dialogue to ensure a thorough evaluation of our work. Your insights are invaluable to us, and we are committed to addressing any remaining issues promptly and thoughtfully.
> > >
> > > Thank you again for your time, effort, and constructive engagement.
> > >
> > > Best regards,
> > >
> > > The Authors

---

> > > > ### Author Response · Authors · 2024-12-03
> > > > **Thank you!**
> > > >
> > > > Dear Reviewer B3ze,
> > > >
> > > > Thank you for your feedback and for engaging with our rebuttal. We're glad to hear that most of your concerns have been addressed and that your impression of the paper remains positive.
> > > >
> > > > As the discussion period ends today, we wanted to highlight our latest zero-shot results for dense models with width-pruning and additional speedup and memory savings comparisons. Below is a summary of our key additions:
> > > >
> > > >   1. **Width-Pruning Results**: Per the reviewer’s suggestion, we included results for width-pruning as a dense baseline. Our updated zero-shot results ([link](https://openreview.net/forum?id=lqHv6dxBkj&noteId=LfxEgmyAEQ)) now include comparisons with Extended SR-STE, Depth-Pruning, and Width-Pruning. These results indicate that our method (SLoPe) outperforms **Extended SR-STE**, **Depth-Pruning**, and **Width-Pruning** on 6, 7, and 7 (including MathQA where both methods achieve the same performance) out of 8 modern zero-shot benchmarks, respectively.
> > > >
> > > >  2. **Speedup and Memory Savings Comparison**: We added an apples-to-apples comparison of speedup and memory savings between SLoPe and Extended SR-STE ([link](https://openreview.net/forum?id=lqHv6dxBkj&noteId=Sii4461Xdc)).
> > > >
> > > > We hope that our effort in thoroughly addressing all questions will be meaningfully reflected in your final assessment of our work.
> > > >
> > > > Thank you once again for your time and valuable input.
> > > >
> > > > Best regards,
> > > >
> > > > The Authors

---

### Official Review · Reviewer_NKfR · 2024-10-28

**Soundness:** 2
**Presentation:** 3
**Contribution:** 2
**Rating:** 5
**Confidence:** 4

**Summary:**

This paper introduces double-pruned sparsity, a technique that applies two rounds of N:M sparsity to the weight matrices of a neural network: one for accelerated forward pass and another on the transpose of the already N:M sparse matrix for an accelerated backward pass. The proposed method is combined with LoRA-tuning in the final 0.01 iterations of pre-training resulting in the method termed Double-Pruned Sparse Plus Lazy Low-rank Adapter Pre-training (SLoPE). The authors use efficient tiling of upsample tensors and kernel fusion to reduce overhead, resulting in further boost in training and inference speed. They show that their approach results in speedup and memory reduction of pre-training LLMs of various sizes. They also show their method improves zero-shot accuracy of GPT2-small compared to a sparse pre-training baseline.

**Strengths:**

1-The proposed double-pruned N:M sparsity for both forward and backward acceleration offers a novel perspective on N:M
sparsity, improving flexibility compared to existing transposable N:M masks.

2-The tiling and kernel fusion solution that integrates LoRAs with sparse weights for enhanced efficiency is interesting.
The authors effectively demonstrate the acceleration and memory reduction benefits of their approach across various models, outperforming FST [1].

3- The method achieves performance surpassing SR-RTE[2] and approaches that of a densely pre-trained model in tasks like MMLU, Arch Challenge, and OpenQA.

[1] Accelerating Transformer Pre-Training with 2:4 Sparsity, Hu et all, ICML 2024

[2] Learning N:M Fine-grained Structured Sparse Neural Networks from Scratch, Zhou et. all, ICLR 2021

**Weaknesses:**

1- The most significant weakness is the limited evaluation of how the proposed sparsity impacts performance. The sparse pre-training method negatively affects performance, and the use of a static sparse matrix further limits flexibility. While the authors claim this does not hinder flexibility, more evidence across a broader range of downstream tasks is needed to support this claim. Given the inherent trade-off between speed and performance, the current experiments do not sufficiently demonstrate the method's effectiveness. Table 3 should include more tasks (tasks in FST[1]) to give the reader a clearer understanding of the impact.

2- Even in the limited downstream task evaluation, some important baselines are missing. The authors put FST in parentheses after SR-STE in the text, although these are distinct methods: FST refers to [1] while ST-RTE is [2]. Zero-shot FST results in their own setting on these tasks are not reported. Additionally, while comparing their method to a larger dense model may be unfair, it would be informative to see how a smaller dense model performs without sparse pre-training.

3- The impact of LoRA fine-tuning in the final iterations appears minimal. The benchmarks (MMLU, Arch Challenge, and OpenQA) are not reported for the LoRA ablations. Since a major contribution of this work is the tiling and kernel fusion solution designed to mitigate issues caused by adding LoRAs, it is concerning that adding LoRAs seems to have little effect, especially based on the current results.

[1] Accelerating Transformer Pre-Training with 2:4 Sparsity, Hu et. all, ICML 2024

[2] Learning N:M Fine-grained Structured Sparse Neural Networks from Scratch, Zhou et. all, ICLR 2021

**Questions:**

1-The authors show that their method is faster than FST during pre-training, which is expected since FST dynamically searches for a mask, whereas their method uses a static one, making the training column in Table 1 clear. However, the inference column of Table 1 suggests that their method is also faster than FST for inference. All sparse pre-training methods use the full model for inference. Does this imply that their method uses the sparse matrix for inference (e.g., during zero-shot evaluations)? Then what is the point of sparse pre-training? In the table caption, it’s mentioned that "the lack of inference speedup in FST is due to the final dense pre-training during the final iterations, resulting in a dense model for inference." Could the authors clarify if you have included fine-tuning on a downstream task as inference? and how are the SLOPE values in the inference columns calculated?

2-Is flash attention used for the baselines as well (line 312)?

**Details Of Ethics Concerns:**

No ethical issues were found.

---

> ### Author Response · Authors · 2024-11-17
> **Part 1/2: Broader Evaluation, LoRA Impact, and Clarifications on Sparse Pre-Training and Inference Speedup**
>
> # Part 1 / 2
>
> We thank the reviewer for their detailed feedback and for pointing out areas where additional clarification and evaluation are needed. Below, we address each of your questions and concerns.
>
> ---
>
> # Evaluation of Sparsity Impact and Flexibility
>
> Thank you for highlighting the importance of a broader evaluation. To address this, we have included results on additional zero-shot downstream teaks using the GLUE benchmark, consistent with the FST paper. These evaluations were conducted using the [Language Modeling Harness](https://github.com/EleutherAI/lm-evaluation-harness/blob/main/lm_eval/tasks/glue/README.md) and the results are summarized in the table below. We are also in contact with the authors of the FST paper to obtain their fine-tuning scripts and will incorporate fine-tuning results as soon as they are available.
>
> Our results demonstrate that SLoPe outperforms Extended SR-STE across various tasks on average, achieving a balance between speed and performance while avoiding the drawbacks of dynamic mask updates. We hope these additional results strengthen the evidence for the effectiveness of our approach across a broader range of tasks.
>
> | Model          | Sparsity       | Method | LoRA (r) | CoLA | MNLI | MNLI-Mismatch | MRPC  | QNLI | QQP  | RTE  | SST2 | WNLI | Average |
> |----------------|----------------|--------|----------|------|------|---------------|-------|------|------|------|------|------|---------|
> | GPT2-Small     | Dense         | -      | r = 0    | 0    | 32.4 | 33.2          | 66.9  | 50.3 | 51.8 | 49.8 | 59.3 | 45.1 | 43.2    |
> | GPT2-Small     | 2:4           | SLoPe  | r = 0    | 0    | 34.3 | 34.0          | 72.5  | 50.0 | 48.5 | 50.0 | 52.3 | 44.0 | 42.84   |
> | GPT2-Small     | 2:4           | SLoPe  | r = 0.05%| 0    | 34.3 | 34.1          | 72.6  | 49.8 | 48.8 | 50.9 | 52.3 | 43.7 | 42.94   |
> | GPT2-Small     | 2:4           | SLoPe  | r = 2.1% | 0    | 34.3 | 34.0          | 71.6  | 50.0 | 49.0 | 52.0 | 52.6 | 45.1 | 43.18   |
> | GPT2-Small     | 2:4           | Extended SR-STE | r = 0    | 0    | 33.6 | 33.9          | 57.1  | 50.7 | 50.4 | 55.2 | 54.7 | 46.5 | 42.46   |
> | GPT2-Small     | 2:4           | Extended SR-STE | r = 0.05%| 0    | 33.1 | 33.6          | 57.9  | 51.0 | 50.5 | 55.4 | 55.0 | 46.5 | 42.55   |
> | GPT2-Small     | 2:4           | Extended SR-STE | r = 2.1% | 0    | 33.3 | 33.5          | 58.2  | 51.0 | 50.5 | 55.2 | 55.2 | 46.5 | 42.6    |
>
>
> Regarding the use of static sparsity, we understand the concern about flexibility. However, dynamic sparsity methods, as demonstrated in SR-STE [1], can introduce instability due to frequent mask updates, potentially impacting training. We have elaborated on these trade-offs in Paper-Appendix-A-Page-15.
>
> ---
>
> # Clarification on Baselines
>
> We apologize for any confusion regarding the terminology. SR-STE is limited to SGD optimizer. The FST paper extends SR-STE for optimizers other than SGD, and we are using that extension in our experiments. Therefore, we have used these terms interchangeably in our accuracy experiments. Additionally, FST provides a fast implementation of the extended optimization method in PyTorch, which we have compared against in our speedup and memory reduction experiments.
>
> ---

---

> ### Author Response · Authors · 2024-11-17
> **Part 2/2: Broader Evaluation, LoRA Impact, and Clarifications on Sparse Pre-Training and Inference Speedup**
>
> #  Comparison with Smaller Dense Models
>
> In response to your suggestion, we have added zero-shot GLUE results and trained a dense GPT model with half the layers of GPT-2 Small (GPT-2 Half) to compare performance with sparse models at equivalent parameter count. The table below summarizes the results, showing that SLoPe, especially with low-rank adapters, consistently outperforms both GPT-2 Half and Extended SR-STE across tasks.
>
>
> | Model          | Sparsity       | Method | LoRA (r) | CoLA  | MNLI  | MNLI-Mismatch | MRPC  | QNLI  | QQP  | RTE   | SST2  | WNLI  | Average       |
> |----------------|----------------|--------|----------|-------|-------|---------------|-------|-------|------|-------|-------|-------|---------------|
> | GPT2-Half      | Dense          | -      | r = 0    | 0.00  | 33.90 | 33.80         | 53.60 | 51.10 | 47.70| 56.70 | 50.60 | 42.30 | 41.08         |
>
> | Model          | Sparsity       | Method | LoRA (r) | MMLU  | Arch Challenge | Open Book QA | Average |
> |----------------|----------------|--------|----------|-------|----------------|--------------|---------|
> | GPT2-Small     | Dense          | -      | r = 0    | 22.9  | 20.7           | 16.2         | 19.94   |
> | GPT2-Small     | 2:4            | SLoPe  | r = 0    | 23.0  | 19.3           | 16.0         | 19.43   |
> | GPT2-Small     | 2:4            | SLoPe  | r = 0.05%| 23.0  | 19.4           | 16.2         | 19.53   |
> | GPT2-Small     | 2:4            | SLoPe  | r = 2.1% | 23.0  | 19.3           | 16.4         | 19.57   |
> | GPT2-Small     | 2:4            | Extended SR-STE | r = 0    | 24.1  | 18.3           | 12.6         | 18.33   |
> | GPT2-Small     | 2:4            | Extended SR-STE | r = 0.05%| 24.1  | 18.4           | 14.2         | 18.90   |
> | GPT2-Small     | 2:4            | Extended SR-STE | r = 2.1% | 24.2  | 18.3           | 14.2         | 18.90   |
> | GPT2-Half      | Dense          | -      | r = 0    | 22.9  | 19.5           | 16.0         | 19.47   |
>
> ---
>
> # LoRA Pretraining Impact and Ablations
>
> Thank you for your observations regarding the impact of low-rank adapters (LoRA).
>
> **LoRA impact during Pre-training** $\rightarrow$ Low-Rank adapters in SLoPe improve accuracy by up to 4.8% on GLUE tasks (CoLA in Table 10 in Page 22) and 0.4% on SQuAD for BERT-Large-Uncased.
>
> **LoRA impact on SR-STE** $\rightarrow$ In addition, our proposed lazy low-rank adapters can boost the accuracy of prior work, such as SR-STE, by up to 1.6% on the OpenBookQA downstream task for GPT-2 Small (Table 3 in Page 9).
>
> **LoRA Ablations** $\rightarrow$ We had the LoRA ablations for MMLU, Arch Challenge, and OpenBookQA in Table 3 (Page 9) of the submitted paper. The Adapter Rank refers to the relative rank of the LoRA compared to the hidden dimension of the model.
>
> ---
>
> # Clarification on Inference Speedup and Sparse Pre-Training
>
> Thank you for your insightful questions.
>
> FST uses a dense fine-tuning step (as an accuracy preserving mechanism), compromising ~17% of the pretraining time (see section 4.4 in FST [2]), resulting in a *dense model for inference* with no inference speedup. In contrast, SLoPe uses lightweight low-rank adapters in the last 1% of pretraining as an accuracy preserving mechanism, maintaining sparse computation benefits. The goal of FST is exclusively to accelerate the training of LLMs, whereas SLoPe aims to accelerate both training and inference.
>
>
> **SLoPe inference computation** $\rightarrow$ During inference, SLoPe does not merge low-rank adapters with the weights. Instead, it employs an efficient sparse matrix-matrix multiplication alongside low-rank adapters, yielding faster inference than dense models. The inference speedups reported for SLoPe were measured using this implementation, whereas the dense inference time for FST reflects its dense fine-tuned model.
>
>
> **SLoPe downstream finetuning** $\rightarrow$ During downstream fine-tuning, SLoPe retains the sparsity mask, updating only the non-zero weights and low-rank adapters, which leads to inference acceleration even after finetuning.
>
> **SLoPe Speedup Measurements** $\rightarrow$ The speedups in the SLoPe column are measured by timing this efficient sparse+low-rank implementation and comparing it with a dense model. For FST, the model at inference is dense because of using dense fine-tuning as their accuracy preserving mechanism, resulting in zero speedup at inference time.
>
> ---
>
> # Clarification on Flash Attention Usage
>
> Thank you for pointing this out. We confirm that Flash Attention 2 was enabled across all baselines, including dense models, FST, and SLoPe. We will clarify this in the revised manuscript to avoid confusion.
>
> ---
>
> [1] Hu et al, “S-STE: Continuous Pruning Function for Efficient 2:4 Sparse Pre-training,” NeurIPS 2024
>
> [2] Hu et al, “Accelerating Transformer Pre-training with 2:4 Sparsity,” ICML 2024

---

> ### Comment · Reviewer_NKfR · 2024-11-23
>
> I sincerely appreciate the authors' efforts to address many of my concerns. However, after carefully reviewing the authors' responses to my comments, as well as those provided to other reviewers, I still have some outstanding concerns:
> 1. Could the authors confirm that all reported SR-STE values in this paper are indeed for FST? What optimizer was used in the experiments? Specifically, does this imply that every contribution and improvement of FST has been accounted for, and the results presented reflect the best possible outcomes for FST? I emphasize this point because there is not a single experiment that is directly comparable between this paper and FST. Either the models, the tasks, or the settings (e.g., zero-shot versus fine-tuning) differ, making it impossible to directly compare this work with what the authors refer to as the latest advancements in sparse pre-training. Moreover, the authors mention that the fine-tuning code of FST is not provided. However, elsewhere, they note that FST performs dense fine-tuning, implying that implementing FST fine-tuning should be straightforward.
>
> 2. Regarding the smaller dense model (Half), it should be designed by reducing the hidden dimensions of each layer, rather than by removing entire blocks. This approach is what was done in FST. Depth pruning generally has a more detrimental effect on model performance compared to width pruning (or reducing the hidden dimensions). For example, this can be observed in Figure 5 of [1]. Besides, the pattern of sparsity introduced in SLoPE resembles width-pruning.
>
> 3. The zero-shot accuracies on GLUE are particularly concerning, especially for SLoPE with r=2.1%
> and baseline SR-STE. These results further highlight the significant impact of a static sparsity mask on performance compared to a dynamic one. While I understand the inherent trade-offs between flexibility, speedup, and fine-tuning accuracy—and that SLoPE aims to balance both pre-training and fine-tuning speed—it is crucial for readers to see accurate results for the smaller dense model. This is necessary to enable a clear assessment of the trade-offs offered by SLoPE.
>
>
> [1] LLM Pruning and Distillation in Practice: The Minitron Approach, Sreenivas et. al, Arxiv 2408.11796

---

> > ### Author Response · Authors · 2024-11-28
> > **[Part 3/3] GPT is not a zero-shot task, static vs. dynamic sparse mask computation dilemma**
> >
> > # GPT is not designed for Classification Tasks!
> >
> > We acknowledge that some tasks show worse performance compared to others, even though the average performance is better. However, we would like to provide additional context about the nature of zero-shot evaluation on GLUE and its limitations.
> >
> > 1. GLUE was not originally designed for zero-shot evaluation, as it primarily targets fine-tuned models for benchmarking. Specifically models like GPT, which are autoregressive, are not inherently well-suited for tasks like classification or question answering compared to BERT-style models. As highlighted in the HuggingFace GitHub discussion [link](https://github.com/huggingface/transformers/issues/9785#issuecomment-768146651): “*BERT has a [CLS] token optimized for classification tasks, whereas GPT processes text left-to-right autoregressively, which makes tasks like GLUE inherently less aligned with its architecture*”.
> > 2. Without fine-tuning, zero-shot results are often noisy and may lead to inconsistent conclusions. We believe that fine-tuning is necessary to obtain more reliable and actionable insights about the performance of different techniques. In this regard, zero-shot results on GLUE may not serve as a productive signal for assessing the overall efficacy of a method. Instead we compare SLoPe and Extended SR-STE across **eight modern downstream tasks** (`Section-3.2-Table-3-Page-9`), specially designed for zero-shot evaluations.
> >
> > ---
> >
> > # Static vs. Dynamic Sparse Mask Computation Dilemma (Pretraining vs. Finetuning)
> >
> > We appreciate this insightful discussion and the opportunity to exchange ideas that could inspire new research directions. Engaging with these questions has been exciting for us as we explore the broader implications of our work.
> >
> > **Dynamic sparse mask is more pivotal for downstream finetuning:** Our hypothesis is that sparse mask recomputation plays a more pivotal role in finetuning than in pretraining. Finetuning typically requires the model to adapt its weights to task-specific features, necessitating more dynamic adjustments in sparsity patterns to align with these nuanced requirements. In contrast, pretraining aims to learn general-purpose representations, where a static or predetermined sparsity pattern can be sufficient and efficient.
> >
> > **Static sparsity is sufficient for pretraining:** Our results are consistent with this hypothesis, demonstrating the effectiveness of static sparsity masks during sparse pretraining. These findings align closely with recent work [1] and [2], which report similar observations. The coherence between our results and the broader body of research reinforces the validity and reliability of our approach.
> >
> > We thank you once again for engaging with our work so thoughtfully, which helped us to think beyond the immediate scope of our study and has highlighted exciting new avenues for future exploration.
> >
> > ---
> >
> > # References
> >
> > [1] Liu et al, “The Unreasonable Effectiveness of Random Pruning: Return of the Most Naive Baseline for Sparse Training,” ICLR 2022
> >
> > [2] Thangarasa et al, SPDF: Sparse Pre-training and Dense Fine-tuning for Large Language Models, UAI 2023

---

> ### Author Response · Authors · 2024-11-28
> **[Part 1/3] Notations, Clarifications about FST, Optimizer, and Finetuning Code**
>
> # Notations
>
> To clarify our response, we have included a set of notations used throughout. This table is also available in the revised manuscript `Appendix-R-Table-14-Page-24`.
>
> $$
> \\require{color}
> \\begin{array}{|l|l|}
> \\hline
> \\rowcolor[gray]{0.9} \\textbf{Term} & \\textbf{Description} \\\\
> \\hline
> \\rowcolor[gray]{0.95} \\text{Sparse Pretraining} & \\text{Common notation used in SLoPe and FST, indicating the use of sparse weights during pretraining} \\\\
> \\hline
> \\rowcolor[gray]{1.0} \\text{Dense Finetuning} & \\text{Notation used in the FST paper, indicating an}\\,  \\underline{\\text{extended pretraining phase}}\\ \\\\
> \\hline
> \\rowcolor[gray]{0.95} \\text{Downstream Finetuning} & \\text{Additional training after pretraining finishes, used to finetune the model for specific downstream tasks} \\\\
> \\hline
> \\rowcolor[gray]{1.0} \\text{FST [1]} & \\text{Sparse Pretraining + Dense Finetuning (which is in fact an extended pretraining)} \\\\
> \\hline
> \\rowcolor[gray]{0.95} \\text{Extended SR-STE} & \\text{Sparse Pretraining with FST algorithm} \\\\
> \\hline
> \\end{array}
> $$
>
> ---
>
>
> # Clarification about SR-STE and FST [1]
>
> We apologize for the confusion and appreciate your comment. To address this, we have revised the manuscript to clearly specify which algorithm (Extended SR-STE or FST) is being used in each comparison. Below, we provide a detailed explanation of these algorithms and the rationale for our choices in the comparisons:
>
> **FST** → We compare SLoPe with FST exclusively for *training speedups and memory savings* (`Table-1-Page-7` and `Table-2-Page-8`). Comparing pretraining quality between SLoPE and FST is less meaningful because the final models produced by these methods differ significantly in the number of parameters (See Table below). Specifically, FST produces a dense model after sparse pretraining and dense finetuning (83% sparse pretraining + 17% dense finetuning), while SLoPe produces a sparse model augmented with lightweight low-rank adaptors (99% sparse pretraining + 1% low-rank adaptation). The final number of parameters after pretraining in SLoPe is approximately **1.48$\times$ smaller** than in the Dense (or FST) model, which makes a direct quality comparison imbalance. That said, we welcome your feedback on how to conduct a meaningful head-to-head quality comparison with FST.
>
> **Extended SR-STE** → We compare SLoPe with Extended SR-STE in terms of model quality, focusing on understanding the dynamics between static and dynamic masking under an equal number of parameters. This allows for a fair, "apple-to-apple" comparison between the methods (iso-params). We refer to this method as "Extended SR-STE" because, while the original SR-STE approach was designed for use with SGD, the FST paper extended it to support other optimizers.
>
>
> $$
> \\require{color}
> \\begin{array}{|l|c|c|}
> \\hline
> \\rowcolor[gray]{0.9} \\textbf{Model} & \\textbf{\\# of Parameters} & \\textbf{Normalized \\# of Parameters vs. Dense \\downarrow} \\\\
> \\hline
> \\rowcolor[gray]{0.95} \\textbf{Dense} & \\textbf{123,563,520} & \\textbf{1.0} \\times\\\\
> \\hline
> \\rowcolor[gray]{1.0} \\textbf{FST (Sparse Pretraining + Dense Finetuning)} & \\textbf{123,563,520}& \\textbf{1.0} \\times\\\\
> \\hline
> \\rowcolor[gray]{0.95} \\textbf{SLoPe (Sparse Pretraining + 2.1\\% LoRA)} & \\textbf{83,771,634}& \\textbf{0.68} \\times\\\\
> \\hline
> \\end{array}
> $$
>
> ---
>
> # Choice of Optimizer
>
> We use ADAMW for both SLoPE and Extended SR-STE. For Extended SR-STE, we strictly follow the mechanism proposed in the FST paper, including the application of masked decay on the gradients. We believe this approach ensures a fair and comprehensive integration of all relevant contributions from the FST paper into our quality comparisons. As mentioned earlier, FST is used exclusively for speedup comparisons, not quality comparisons, to highlight its intended strengths in those aspects. For ease of reference, we added code snippets of our implementation and FST implementation to `Appendix-R-Page-24-25`.
>
>
> ---
>
> # Fine-tuning Code of FST
>
> Thank you for your suggestion. The publicly available [FST codebase](https://github.com/huyz2023/2by4-pretrain) includes implementations for sparse pretraining and dense finetuning (this is extended pretraining, please see the notation table above), which we have adopted in our comparisons. However, the authors have not released the codebase nor the configurations for downstream finetuning on tasks like GLUE, making it challenging for us to reproduce their results in Table 6 at this time. The authors have assured us that they will share the downstream finetuning code after their deadline, and we plan to run the relevant experiments as soon as we gain access to it. Until then, all results we present are based on **zero-shot downstream evaluation**, without any additional downstream finetuning.
>
> ---
>
> # References
>
> [1] Hu et al, “Accelerating Transformer Pre-training with 2:4 Sparsity,” ICML 2024

---

> ### Author Response · Authors · 2024-11-28
> **[Part 2/3] Depth/Width-Pruning, Zero-Shot Accuracies, Five Additional Zero-Shot Results**
>
> # Depth-Pruning vs. Width-Pruning for Reduced-Sized Dense Models
>
> Thank you for bringing this paper to our attention. After carefully reviewing the paper, we would like to clarify that [1] focuses on comparing the quality of depth-pruning vs. width-pruning after full dense pretraining. Specifically, the authors pretrain a dense model of size P and then prune it during finetuning to obtain a reduced-sized model of approximately P/2 for downstream tasks. the model for the downstream tasks to obtain a reduced-sized model of size ~P/2. We will include a detailed discussion to highlight the differences between their method and ours, ensuring that the contributions of both works are properly contextualized.
>
> While the context differs from our focus on sparse pretraining methods, we found your suggestion intriguing and initiated a set of experiments to explore the tradeoffs between these pruning strategies in our setting. We look forward to incorporating these insights into our work and thank you again for the thoughtful suggestion.
>
> **Disclaimer:** We pretrained LLaMA-2-7b and Gemma2-2b and Gemma2-9b models using the [MaxText](https://github.com/AI-Hypercomputer/maxtext) codebase on TPU. This choice was motivated by access to a larger pool of TPU resources, which enabled us to obtain preliminary results more quickly and share them within the discussion period. Please see the configurations for each model in `Appendix-Table-15,16,17-Page-26,27`.
>
>
> **Results:** Preliminary pretraining loss curves suggest no significance difference between depth-pruning and width-pruning during pretraining. Interestingly, in some cases, depth-pruning appears to outperform width-pruning. These early loss curves are included in the `Appendix-Figure-10-Page-28`.
>
> We acknowledge that these are early results, shared to provide insights before the discussion period concludes. Thanks to your comment, we believe that a comprehensive analysis of the impact of depth-pruning vs. width-pruning represents a valuable avenue for future work.
>
> ---
>
> # Zero-shot Accuracies on GLUE
>
> Thank you for reviewing the results. We understand that per-task model quality could differ from one method to another. A couple of clarifications for comparing model quality between models with the same number of parameters:
>
>
> **(1) Zero-Shot GLUE Results:** SLoPe outperforms Extended SR-STE in 3/8 tasks as well as average (SLoPE = 43.18 vs. Extended SR-STE = 42.6).
>
>
>
> **(2) Modern Zero-Shot Benchmarks:** In more recent zero-shot eval benchmarks (MMLU, Arch Challenge, and Open Book QA) Slope outperforms Extended SR-STE in 2/3 tasks as well as the average quality (19.57 vs. 18.90)
>
> $$
> \\require{color}
> \\begin{array}{|c|c|c|c|c|c|c|c|}
> \\hline
> \\rowcolor[gray]{0.9} \\textbf{Model} & \\textbf{Sparsity} & \\textbf{Method} & \\textbf{LoRA (r)} & \\textbf{MMLU} & \\textbf{Arch Challenge} & \\textbf{Open Book QA} & \\textbf{Average} \\\\
> \\hline
> \\rowcolor[gray]{0.95} \\text{GPT2-Small} & \\text{Dense} & - & r = 0 & 22.9 & 20.7 & 16.2 & 19.94 \\\\
> \\hline
> \\rowcolor[gray]{1.0} \\text{GPT2-Small} & \\text{2:4} & \\text{SLoPe} & r = 2.1\\% & 23.0 & \\textbf{19.3} & \\textbf{16.4} & \\textbf{19.57} \\\\
> \\hline
> \\rowcolor[gray]{0.95} \\text{GPT2-Small} & \\text{2:4} & \\text{Extended SR-STE} & r = 2.1\\% & \\textbf{24.2} & 18.3 & 14.2 & 18.90 \\\\
> \\hline
> \\end{array}
> $$
>
> ---
>
> # Additional Zero-Shot Results
>
> To further demonstrate the effectiveness of SLoPe, we conducted additional evaluations on five modern zero-shot benchmark tasks using the [Language Model Evaluation Harness](https://github.com/EleutherAI/lm-evaluation-harness/tree/main).
>
> As shown in the table below, SLoPe consistently outperforms Extended SR-STE (same number of parameters) on four out of five tasks. These results underscore the advantages of our method’s static sparsity masks. This performance gain highlights the robustness and practicality of SLoPe in leveraging sparsity during pretraining across diverse tasks.
>
> Additional results, including evaluations on different configurations and metrics, are provided in the (`Section-3.2-Table-3, Page-9`) for further validation.
>
> $$
> \\require{color}
> \\begin{array}{|c|c|c|c|c|c|c|}
> \\hline
> \\rowcolor[gray]{0.9} \\textbf{Method} & \\textbf{Adapter Rank} & \\textbf{Winogrande \uparrow} & \\textbf{HellaSwag \uparrow} & \\textbf{MathQA \uparrow} & \\textbf{PiQA \uparrow} & \\textbf{Race \uparrow} \\\\
> \\hline
> \\rowcolor[gray]{0.95} \\text{Dense} & - & 50.6 & 28.5 & 21.8 & 59.8 & 28.4 \\\\
> \\hline
> \\rowcolor[gray]{1.0} \\text{SLoPe} & r=2.1\\% & \\textbf{50.8} & \\textbf{27.5} & 20.8 & \\textbf{57.6} & \\textbf{27.2} \\\\
> \\hline
> \\rowcolor[gray]{0.95} \\text{Extended SR-STE} & r=2.1\\% & 47.5 & 26.9 & \\textbf{21.4} & 55.2 & 24.2 \\\\
> \\hline
> \\end{array}
> $$
>
> ---
>
> # References
>
> [1] Sreenivas et. al, “LLM Pruning and Distillation in Practice: The Minitron Approach,” Arxiv 2024

---

> ### Comment · Reviewer_NKfR · 2024-12-01
>
> I thank the authors for their detailed response and the clarifications provided.
> ## Apples and Oranges
> I did not receive a definitive answer to one of my key questions. While the authors argue that comparing the accuracies of SLoPE and FST constitutes an "apples-to-oranges" comparison, I find it inconsistent that much of Table 1 and Table 2 is devoted to comparing the speed-up and memory usage of SLoPE with FST, using values from the dense model resulting from the application of FST. Wouldn’t this also fall under the category of an "apples-to-oranges" comparison?
>
> I would like to clarify my question. I apologize if my original question was not clear:
> **Are the results reported as "Extended SR-STE" identical to those of FST without the full fine-tuning stage, assuming an independent individual were to reproduce the results presented in this paper?**
>
> ## Dense Baseline
>
> I find it perplexing that significant resources were allocated to these additional width-depth sparsity comparison experiments, yet no dense pretraining experiment was conducted using the correct dense baseline. This seems particularly crucial given the role of a dense baseline in contextualizing the sparsity trade-offs of SLoPE. As I said before, the sparsity pattern introduced in SLoPE resembles width-pruning.
>
> While I understand that the discussion period has now concluded and no further experiments can be added, my original point still stands: this paper lacks a proper dense baseline designed by reducing the width of each layer rather than removing entire blocks. This omission limits the ability to fully assess the advantages and trade-offs of SLoPE compared to an appropriate dense baseline.

---

> ### Author Response · Authors · 2024-12-02
> **[Part 1/2] Additional Experiments, Identical Extended SR-STE, Dense Baseline**
>
> Dear Reviewer NKfR,
>
> Thank you for your active engagement during the rebuttal period and your suggestions.
>
> # Apples and Oranges
>
> We compare SLoPe with FST for speedup and memory savings as FST is the only baseline in sparse pretraining that demonstrates practical  acceleration and memory savings on GPUs.
>
> To provide further context, the table below presents a detailed comparison of SLoPe and Extended-SR-STE in terms of speedup and memory savings:
>
> ### < Speedup ($\uparrow$ is better) >
>
> $$
> \\require{color}
> \\begin{array}{|l|l|c|}
> \\hline
> \\rowcolor[gray]{0.9} \\textbf{Model} & \\textbf{Method} & \\textbf{Training Speedup} \\\\
> \\hline
> \\rowcolor[gray]{0.95} \\text{OPT-66B} & \\text{SLoPe} & \\mathbf{1.20} \\\\
> \\hline
> \\rowcolor[gray]{0.95} \\text{OPT-66B} & \\text{Extended SR-STE} & 1.14 \\\\
> \\hdashline
> \\text{OPT-30B} & \\text{SLoPe} & \\mathbf{1.22} \\\\
> \\hline
> \\text{OPT-30B} & \\text{Extended SR-STE} & 1.12 \\\\
> \\hdashline
> \\rowcolor[gray]{0.95} \\text{OPT-13B} & \\text{SLoPe} & \\mathbf{1.25} \\\\
> \\hline
> \\rowcolor[gray]{0.95} \\text{OPT-13B} & \\text{Extended SR-STE} & 1.16 \\\\
> \\hdashline
> \\text{OPT-6.6B} & \\text{SLoPe} & \\mathbf{1.21} \\\\
> \\hline
> \\text{OPT-6.6B} & \\text{Extended SR-STE} & 1.13 \\\\
> \\hdashline
> \\rowcolor[gray]{0.95} \\text{OPT-2.6B} & \\text{SLoPe} & \\mathbf{1.13} \\\\
> \\hline
> \\rowcolor[gray]{0.95} \\text{OPT-2.6B} & \\text{Extended SR-STE} & 1.06 \\\\
> \\hdashline
> \\text{LLaMA-3-8B} & \\text{SLoPe} & \\mathbf{1.16} \\\\
> \\hline
> \\text{LLaMA-3-8B} & \\text{Extended SR-STE} & 1.08 \\\\
> \\hdashline
> \\rowcolor[gray]{0.95} \\text{Mistral-v0.3-7B} & \\text{SLoPe} & \\mathbf{1.15} \\\\
> \\hline
> \\rowcolor[gray]{0.95} \\text{Mistral-v0.3-7B} & \\text{Extended SR-STE} & 1.07 \\\\
> \\hline
> \\end{array}
> $$
>
> ### < Memory Reduction ($\downarrow$ is better) >
>
> $$
> \\require{color}
> \\begin{array}{|l|l|c|}
> \\hline
> \\rowcolor[gray]{0.9} \\textbf{Model} & \\textbf{Method} & \\textbf{Training Memory Reduction} \\\\
> \\hline
> \\rowcolor[gray]{0.95} \\text{OPT-66B} & \\text{SLoPe} & \\mathbf{0.67} \\\\
> \\hline
> \\rowcolor[gray]{0.95} \\text{OPT-66B} & \\text{Extended SR-STE} & 1.27 \\\\
> \\hdashline
> \\text{OPT-30B} & \\text{SLoPe} & \\mathbf{0.67} \\\\
> \\hline
> \\text{OPT-30B} & \\text{Extended SR-STE} & 1.20 \\\\
> \\hdashline
> \\rowcolor[gray]{0.95} \\text{OPT-13B} & \\text{SLoPe} & \\mathbf{0.68} \\\\
> \\hline
> \\rowcolor[gray]{0.95} \\text{OPT-13B} & \\text{Extended SR-STE} & 1.19 \\\\
> \\hdashline
> \\text{OPT-6.6B} & \\text{SLoPe} & \\mathbf{0.68} \\\\
> \\hline
> \\text{OPT-6.6B} & \\text{Extended SR-STE} & 1.22 \\\\
> \\hdashline
> \\rowcolor[gray]{0.95} \\text{OPT-2.6B} & \\text{SLoPe} & \\mathbf{0.67} \\\\
> \\hline
> \\rowcolor[gray]{0.95} \\text{OPT-2.6B} & \\text{Extended SR-STE} & 1.20 \\\\
> \\hdashline
> \\text{LLaMA-3-8B} & \\text{SLoPe} & \\mathbf{0.63} \\\\
> \\hline
> \\text{LLaMA-3-8B} & \\text{Extended SR-STE} & 1.17 \\\\
> \\hdashline
> \\rowcolor[gray]{0.95} \\text{Mistral-v0.3-7B} & \\text{SLoPe} & \\mathbf{0.68} \\\\
> \\hline
> \\rowcolor[gray]{0.95} \\text{Mistral-v0.3-7B} & \\text{Extended SR-STE} & 1.16 \\\\
> \\hline
> \\end{array}
> $$
>
> $^*$ Values larger than 1.00 in the memory reduction table indicate memory overhead. Extended-SR-STE (and FST during their sparse pretraining) stores the dense weight, in addition to the sparse weight and its transpose, leading to a memory overhead in comparison to the dense baseline.
>
> $^*$ The inference speedup and memory reduction of Extended-SR-STE and SLoPe are identical, when using the efficient SLoPe tiling and low-rank fusion implementation.
>
> ---
>
> # Identical Extended SR-STE
>
> > Are the results reported as "Extended SR-STE" identical to those of FST without the full fine-tuning stage, assuming an independent individual were to reproduce the results presented in this paper?
>
> Yes.
>
> We apologize if our answer was not clear enough in our previous response. As we mentioned:
>
> - [Part 1/3] Notation Table clearly states that Extended-SR-STE is the sparse pretraining part of the FST algorithm.
> $$
> \\require{color}
> \\begin{array}{|l|l|}
> \\hline
> \\rowcolor[gray]{0.9} \\textbf{Term} & \\textbf{Description} \\\\
> \\hline
> \\rowcolor[gray]{0.95} \\text{Extended SR-STE} & \\text{Sparse Pretraining with FST algorithm} \\\\
> \\hline
> \\end{array}
> $$
> - [Part 1/3] In the **Choice of Optimizer** section we mentioned that we provided the code snippet for FST and Extended SR-STE (our implementation) that shows we use the same implementation.
>
> ---

---

> ### Author Response · Authors · 2024-12-02
> **[Part 2/2] Additional Experiments, Identical Extended SR-STE, Dense Baseline**
>
> # Dense Baseline
> We want to reassure you that we’re doing our best to address all comments thoroughly and on time. The additional experiments during the discussion period were possible because we had access to *dedicated* TPU resources, which allowed us to provide results quickly. Unfortunately, the GPT experiments are taking longer since they’re running on a shared GPU cluster (64×V100). We’re working hard to get these results ready by the deadline and appreciate the reviewer’s understanding of these practical limitations.
>
> On the comparison between depth-pruning and width-pruning, we were intrigued by your comment. In our experiment across three models, we observed that depth-pruning and width-pruning consistently resulted in identical loss values during pre-training. This was true across different model architectures, which makes us believe the two approaches lead to equivalent outcomes in this context. If there’s something we might be overlooking or if you have reasons to expect different results, we’d love to hear your perspective and explore this further.

---

> > ### Author Response · Authors · 2024-12-03
> > **Width-Pruning Dense Baseline Zero-shot Results**
> >
> > Dear Reviewer NKfR,
> >
> > Thank you for your valuable suggestion to include width-pruning results as the “*correct dense baseline*”. We are pleased to share that we have conducted the requested experiments, and the results are fresh from the oven! As you pointed out, width-pruning serves as an important baseline, and we appreciate your insight, which has helped us to further strengthen our work.
> >
> > In our new experiments, we evaluated models pruned via width pruning, where the number of parameters in each pruned model is halved while keeping the total number of transformer layers intact. Below are the new results:
> >
> > ### Accuracy Results for Modern Zero-shot Benchmarks (LLM Harness)
> >
> > The results of our experiments are summarized in the table below. The results demonstrate that SLoPe outperforms Extended SR-STE, Width-Pruning, and Depth-Pruning on 6, 7, and 7 (including MathQA where both methods achieve the same performance) out of 8 modern zero-shot benchmarks, respectively.
> >
> > $$
> > \\require{color}
> > \\begin{array}{|l|c|c|c|}
> > \\hline
> > \\rowcolor[gray]{0.9} \\textbf{Method} & \\textbf{Adapter Rank} & \\textbf{MMLU \uparrow} & \\textbf{Arch Challenge \uparrow} & \\textbf{Open Book QA \uparrow} \\\\
> > \\hline
> > \\rowcolor[gray]{0.95} \\text{Dense} & - & 22.9 & 20.7 & 16.2 \\\\
> > \\hline
> > \\rowcolor[gray]{1.0} \\text{SLoPe} & r=2.1\\% & 23.0 & 19.3 & \\textbf{16.4} \\\\
> > \\hdashline
> > \\rowcolor[gray]{0.95} \\text{Extended SR-STE} & r=2.1\\% & \\textbf{24.2} & 18.3 & 14.2 \\\\
> > \\hline
> > \\rowcolor[gray]{1.0} \\text{Width-Pruning} & – & 22.9 & 17.7 & 14.0 \\\\
> > \\hdashline
> > \\rowcolor[gray]{0.95} \\text{Depth-Pruning} & – & 22.9 & \\textbf{19.5} & 16.0 \\\\
> > \\hline
> > \\end{array}
> > $$
> >
> > $$
> > \\require{color}
> > \\begin{array}{|c|c|c|c|c|c|c|}
> > \\hline
> > \\rowcolor[gray]{0.9} \\textbf{Method} & \\textbf{Adapter Rank} & \\textbf{Winogrande \uparrow} & \\textbf{HellaSwag \uparrow} & \\textbf{MathQA \uparrow} & \\textbf{PiQA \uparrow} & \\textbf{Race \uparrow} \\\\
> > \\hline
> > \\rowcolor[gray]{0.95} \\text{Dense} & - & 50.6 & 28.5 & 21.8 & 59.8 & 28.4 \\\\
> > \\hline
> > \\rowcolor[gray]{1.0} \\text{SLoPe} & r=2.1\\% & \\textbf{50.8} & \\textbf{27.5} & 20.8 & \\textbf{57.6} & \textbf{27.2} \\\\
> > \\hdashline
> > \\rowcolor[gray]{0.95} \\text{Extended SR-STE} & r=2.1\\% & 47.5 & 26.9 & \\textbf{21.4} & 55.2 & 24.2 \\\\
> > \\hline
> > \\rowcolor[gray]{1.0} \\text{Width-Pruning} & –  & 50.6 & 26.9 & 21.1 & 56.0 & 26.0 \\\\
> > \\hdashline
> > \\rowcolor[gray]{0.95} \\text{Depth-Pruning} & – & 49.0 & 27.4 & 20.8 & 57.3 & 26.8 \\\\
> > \\hline
> > \\end{array}
> > $$
> >
> > $^*$ Bold values indicate the highest accuracy achieved among the compressed models.
> >
> > ---
> >
> > These additional evaluations demonstrate the robustness of SLoPe and reinforce its effectiveness. We kindly request you to consider revising your assessment in light of these new findings.
> >
> > Your feedback has been instrumental in improving the rigor of our work, and we are committed to including this discussion and the results in the final version of our manuscript.
> >
> > Thank you again for your thoughtful feedback and engagement with our paper.
> >
> > Best regards,
> >
> > The Authors

---

### Official Review · Reviewer_ywJx · 2024-11-04

**Soundness:** 3
**Presentation:** 3
**Contribution:** 3
**Rating:** 6
**Confidence:** 2

**Summary:**

This paper introduces SLOPE, a novel pretraining method aimed at improving the efficiency of large language models (LLMs) by combining sparsity and low-rank approximations. SLOPE employs double-pruning techniques—applying N:M sparsity masks during both forward and backward passes—and introduces lazy low-rank adapters in the final training stages. The double-pruning approach reduces memory and computational overheads, while the lazy low-rank adapters help maintain model accuracy by mitigating the quality loss from sparsity.

**Strengths:**

1. This is a technical solid article that designs optimized CUDA kernels that jointly optimize Nvidia 2:4 sparse kernels and low-rank calls through efficient tiling and scheduling.

2. SLOPE yields up to 1.25x faster training and 1.54x faster inference on models with billions of parameters, with a memory reduction of up to 63% during training.

**Weaknesses:**

1. Structured and semi-structured sparsity with low-rank adapters, are not new concepts in the literature (e.g., see Losparse [1]).

2. This article lacks an introduction to related work and baseline methods, such as FST. Readers unfamiliar with the field may find it challenging to follow the content.

3. Could you provide results on additional zero-shot datasets in GLUE beyond MMLU, ARC-c, and OpenBookQA for GPT-2 when comparing SR-STE?


[1] Li, Yixiao, et al. "Losparse: Structured compression of large language models based on low-rank and sparse approximation." In International Conference on Machine Learning, pp. 20336-20350. PMLR, 2023.

Minor:
1. L 430 typo of "accuracy"

2. The bold values in the Tables are misleading:

Table 1: Why are some values of FST also bolded?

Table 3: Why are the values of SR-STE not bolded on the MMLU dataset?

**Questions:**

Please see the weakness.

---

> ### Author Response · Authors · 2024-11-17
> **Expanding Related Work, Zero-Shot GLUE Results, and Resolving Formatting Issues**
>
> We thank you for your feedback and positive assessment of our work. Below, we address your specific questions in detail.
>
> ---
>
> # Novelty in Combining Sparsity and Low-Rank Adapters for Pretraining
>
> We appreciate your recognition of the importance of sparsity and low-rank adapters in prior work, such as LoSparse [1]. While these techniques have indeed been studied previously, our work introduces a unique approach by applying sparsity and low-rank adapters in both the *forward* and *backward* passes (doubly pruned), accelerating not only model training but also inference. This algorithmic contribution is further complemented by the implementation of new tiling and low-rank inference CUDA kernels, enabling more efficient training and inference.
>
> Additionally, LoSparse [1] uses low-rank adapters throughout the entire fine-tuning process, which limits the potential speedup and incurs additional computational overhead. In contrast, our approach strategically integrates lazy low-rank adapters exclusively during the final 1\% of pretraining, effectively reducing overhead while maintaining their expressive capacity. This combination optimizes both efficiency and performance during pretraining, addressing a key gap in existing techniques.
>
> We have already cited LoSparse in the paper, but in response to your feedback, we have included additional details in Appendix-Section-N (Page 23).
>
> ---
>
> # Clarification on Related Work and Baseline Coverage
>
> We thank the reviewer for highlighting the need for a more detailed discussion of related work and baseline methods. To address this, we have expanded the related work section in Section N of the appendix (Page 22) to include a comprehensive discussion of baseline methods, such as FST, and other relevant studies. Additionally, we have highlighted key related works in the introduction. We hope this combination offers a balanced overview, but we are open to further expanding these sections to improve clarity for readers unfamiliar with the field.
>
> ---
>
> # Additional Zero-Shot GLUE Results
>
> Thank you for the suggestion to include results on additional zero-shot datasets in GLUE. We have added an extended zero-shot evaluation using the GLUE benchmark with the [Language Modeling Harness](https://github.com/EleutherAI/lm-evaluation-harness/blob/main/lm_eval/tasks/glue/README.md). The table below presents the performance of dense, SR-STE, and SLoPe training schemes, showing that SLoPe outperforms Extended SR-STE on average.
>
> Additionally, we have contacted the authors of the FST paper to obtain their fine-tuning scripts and will incorporate fine-tuning GLUE results once available. We hope these additional results provide more comprehensive comparisons and address your concerns.
>
> We have included this table in the revised manuscript, Appendix-Section-Q-Table-13 (Page 24).
>
>
> | Model          | Sparsity       | Method | LoRA (r) | CoLA | MNLI | MNLI-Mismatch | MRPC  | QNLI | QQP  | RTE  | SST2 | WNLI | Average |
> |----------------|----------------|--------|----------|------|------|---------------|-------|------|------|------|------|------|---------|
> | GPT2-Small     | Dense         | -      | r = 0    | 0    | 32.4 | 33.2          | 66.9  | 50.3 | 51.8 | 49.8 | 59.3 | 45.1 | 43.2    |
> | GPT2-Small     | 2:4           | SLoPe  | r = 0    | 0    | 34.3 | 34.0          | 72.5  | 50.0 | 48.5 | 50.0 | 52.3 | 44.0 | 42.84   |
> | GPT2-Small     | 2:4           | SLoPe  | r = 0.05%| 0    | 34.3 | 34.1          | 72.6  | 49.8 | 48.8 | 50.9 | 52.3 | 43.7 | 42.94   |
> | GPT2-Small     | 2:4           | SLoPe  | r = 2.1% | 0    | 34.3 | 34.0          | 71.6  | 50.0 | 49.0 | 52.0 | 52.6 | 45.1 | 43.18   |
> | GPT2-Small     | 2:4           | Extended SR-STE | r = 0    | 0    | 33.6 | 33.9          | 57.1  | 50.7 | 50.4 | 55.2 | 54.7 | 46.5 | 42.46   |
> | GPT2-Small     | 2:4           | Extended SR-STE | r = 0.05%| 0    | 33.1 | 33.6          | 57.9  | 51.0 | 50.5 | 55.4 | 55.0 | 46.5 | 42.55   |
> | GPT2-Small     | 2:4           | Extended SR-STE | r = 2.1% | 0    | 33.3 | 33.5          | 58.2  | 51.0 | 50.5 | 55.2 | 55.2 | 46.5 | 42.6    |
>
> ---
>
> # Addressing Typos and Formatting Issues
>
> Thank you for highlighting the typos and formatting inconsistencies, particularly the bold value errors in the tables. We apologize for these mistakes and have corrected them in the revised manuscript.
>
> ---
>
> [1] Li, Yixiao, et al. "LoSparse: Structured compression of large language models based on low-rank and sparse approximation." ICML 2023

---

> ### Author Response · Authors · 2024-11-28
> **Gentle Reminder, Clarifications, Additional Zero-shot Results,**
>
> Dear Reviewer ywJx,
>
> Thank you once again for your time and thoughtful feedback on our submission. As we approach the end of the discussion period, we wanted to gently remind you that we have updated the manuscript to address the questions raised. Specifically, we have revised the paper to clarify the notations and provided a fair comparison between FST and SLoPe.
>
> Please refer to our three-part response to Reviewer NKfR ([Part1](https://openreview.net/forum?id=lqHv6dxBkj&noteId=dAl1Bi4ccu), [Part2](https://openreview.net/forum?id=lqHv6dxBkj&noteId=IXrHn9aK3I), and [Part3](https://openreview.net/forum?id=lqHv6dxBkj&noteId=kOvkHigUk1)) for details.
>
> # Notations
>
> To clarify our response, we have included a set of notations used throughout. This table is also available in the revised manuscript `Appendix-R-Table-14-Page-24`.
>
> $$
> \\require{color}
> \\begin{array}{|l|l|}
> \\hline
> \\rowcolor[gray]{0.9} \\textbf{Term} & \\textbf{Description} \\\\
> \\hline
> \\rowcolor[gray]{0.95} \\text{Sparse Pretraining} & \\text{Common notation used in SLoPe and FST, indicating the use of sparse weights during pretraining} \\\\
> \\hline
> \\rowcolor[gray]{1.0} \\text{Dense Finetuning} & \\text{Notation used in the FST paper, indicating an}\\,  \\underline{\\text{extended pretraining phase}}\\ \\\\
> \\hline
> \\rowcolor[gray]{0.95} \\text{Downstream Finetuning} & \\text{Additional training after pretraining finishes, used to finetune the model for specific downstream tasks} \\\\
> \\hline
> \\rowcolor[gray]{1.0} \\text{FST [1]} & \\text{Sparse Pretraining + Dense Finetuning (which is in fact an extended pretraining)} \\\\
> \\hline
> \\rowcolor[gray]{0.95} \\text{Extended SR-STE} & \\text{Sparse Pretraining with FST algorithm} \\\\
> \\hline
> \\end{array}
> $$
>
> ---
>
> # Eight Modern Zero-shot Benchmarks
>
> We also added eight modern zero-shot benchmarks from [Language Model Evaluation Harness](https://github.com/EleutherAI/lm-evaluation-harness/tree/main). The tables below summarize the results. SLoPe outperforms Extended SR-STE (same number of parameters) on **six out of eight tasks**.
>
> ---
>
> ## Initial zero-shot benchmarks
>
> $$
> \\require{color}
> \\begin{array}{|l|c|c|c|}
> \\hline
> \\rowcolor[gray]{0.9} \\textbf{Method} & \\textbf{Adapter Rank} & \\textbf{MMLU \uparrow} & \\textbf{Arch Challenge \uparrow} & \\textbf{Open Book QA \uparrow} \\\\
> \\hline
> \\rowcolor[gray]{0.95} \\text{Dense} & - & 22.9 & 20.7 & 16.2 \\\\
> \\hline
> \\rowcolor[gray]{1.0} \\text{SLoPe} & r=2.1\\% & 23.0 & \\textbf{19.3} & \\textbf{16.4} \\\\
> \\hline
> \\rowcolor[gray]{0.95} \\text{Extended SR-STE} & r=2.1\\% & \\textbf{24.2} & 18.3 & 14.2 \\\\
> \\hline
> \\end{array}
> $$
>
> ---
>
> ## Additional Zero-shot Results
>
> $$
> \\require{color}
> \\begin{array}{|l|c|c|c|c|c|c|c|c|c|}
> \\hline
> \\rowcolor[gray]{0.9} \\textbf{Method} & \\textbf{Adapter Rank} & \\textbf{Winogrande \uparrow} & \\textbf{HellaSwag \uparrow} & \\textbf{MathQA \uparrow} & \\textbf{PiQA \uparrow} & \\textbf{Race \uparrow}  \\\\
> \\hline
> \\rowcolor[gray]{0.95} \\text{Dense} & - & 50.6 & 28.5 & 21.8 & 59.8 & 28.4 \\\\
> \\hline
> \\rowcolor[gray]{1.0} \\text{SLoPe} & r=2.1\\% & \\textbf{50.8} & \\textbf{27.5} & 20.8 & \\textbf{57.6} & \\textbf{27.2} \\\\
> \\hline
> \\rowcolor[gray]{0.95} \\text{Extended SR-STE} & r=2.1\\% & 47.5 & 26.9 & \\textbf{21.4} & 55.2 & 24.2 \\\\
> \\hline
> \\end{array}
> $$
>
> ---
>
> We would greatly appreciate your thoughts on the revisions, as your feedback is invaluable to ensuring the work is evaluated fairly and comprehensively.
> Thank you for your attention during this process.
>
>
> Best regards,
>
> The Authors

---

> ### Comment · Reviewer_ywJx · 2024-12-03
> **Reply by Reviewer**
>
> Thanks for the detailed reply from the authors. Most of my concerns have been addressed. I'd like to retain my score as positive.

---

> > ### Author Response · Authors · 2024-12-03
> > **Thank you!**
> >
> > Dear Reviewer ywJx,
> >
> > Thank you for your feedback and for engaging with our rebuttal. We're glad to hear that most of your concerns have been addressed and that your impression of the paper remains positive.
> >
> > As the discussion period ends today, we wanted to highlight our latest zero-shot results for dense models with width-pruning and additional speedup and memory savings comparisons. Below is a summary of our key additions:
> >
> >   1. **Width-Pruning Results**: Per the reviewer’s suggestion, we included results for width-pruning as a dense baseline. Our updated zero-shot results ([link](https://openreview.net/forum?id=lqHv6dxBkj&noteId=LfxEgmyAEQ)) now include comparisons with Extended SR-STE, Depth-Pruning, and Width-Pruning. These results indicate that our method (SLoPe) outperforms **Extended SR-STE**, **Depth-Pruning**, and **Width-Pruning** on 6, 7, and 7 (including MathQA where both methods achieve the same performance) out of 8 modern zero-shot benchmarks, respectively.
> >
> >  2. **Speedup and Memory Savings Comparison**: We added an apples-to-apples comparison of speedup and memory savings between SLoPe and Extended SR-STE ([link](https://openreview.net/forum?id=lqHv6dxBkj&noteId=Sii4461Xdc)).
> >
> > We hope that our effort in thoroughly addressing all questions will be meaningfully reflected in your final assessment of our work.
> >
> > Thank you once again for your time and valuable input.
> >
> > Best regards,
> >
> > The Authors

---

### Author Response · Authors · 2024-11-17
**Thank you!**

Dear Area Chairs and Reviewers,

We are grateful for the time you have dedicated to reviewing our manuscript and for your insightful feedback. We appreciate the positive remarks from the reviewers, praising SLoPe for its (1) novelty and a technical solidity with optimized CUDA kernels with efficient tiling and scheduling [Reviewer ywjx](https://openreview.net/forum?id=lqHv6dxBkj&noteId=f2zY0XLoGq), (2) surpassing other sparse and dense methods in accuracy and effective demonstration of acceleration and memory reduction [Reviewer NKfR](https://openreview.net/forum?id=lqHv6dxBkj&noteId=1lfuUA3d4g), (3) significant speedup in both pretraining and inference with a well-written and easy to follow manuscript [Reviewer B3ze](https://openreview.net/forum?id=lqHv6dxBkj&noteId=XBJWBmV8Ay). Your comments have been instrumental in refining our work. Consequently, we have added further clarifications and additional results, which are summarized below:

- [Rebuttal-A1](https://openreview.net/forum?id=lqHv6dxBkj&noteId=MxWgE5lxsL), [Rebuttal-A2](https://openreview.net/forum?id=lqHv6dxBkj&noteId=Qykay4JQDw): Testing SLoPe and other benchmarks on the GLUE zero-shot downstream task.

- [Rebuttal-A2](https://openreview.net/forum?id=lqHv6dxBkj&noteId=ND6rig1pY3): Pretraining a fully dense model with the same size as the sparse models and reporting GLUE, MMLU, Arch Challenge, and OpenBookQA to further clarify the effectiveness of SLoPe.

We are confident that we have responded to each of the reviewers' comments individually. We look forward to engaging in a constructive discussion during the author response period.

Best regards,

Authors

---

### Comment · Area_Chair_KaQF · 2024-11-23
**Follow-Up Discussion on Author Feedback**

Dear PC Members,

Thank you for your valuable comments during the review period, which raised many interesting and insightful questions. The authors have now posted their feedback, and I encourage you to review their responses and engage in further discussion if necessary.

I understand that you may have a busy schedule, but your additional input is highly appreciated, particularly for papers that are initially on the borderline. Your contributions are crucial in ensuring a fair and well-rounded decision-making process.

Thank you once again for your continued support and dedication to ICLR.

Best regards,
AC

---

### Meta-Review · Area_Chair_KaQF · 2024-12-20

**Metareview:**

This paper introduces a novel pruning technique based on double-direction masks, which is further combined with LoRA in the final stages, resulting in the Double-Pruned Sparse Plus Lazy Low-rank Adapter Pre-training (SLoPE). The reviewers recognized the solid technical contributions of this work. However, there are lingering concerns about the experiments, particularly regarding the use of a dense training baseline and static masks. While some of the reviewers' questions were addressed during the rebuttal, the responses only partially resolved their concerns. As a result, the overall scores are generally positive, though one reviewer with a score of 5 noted that they might increase their score if other reviewers had no remaining doubts about the experiments.

Overall, I believe this is a borderline paper: it offers technical contributions, but they are not highly significant. I suggest acceptance, but I am open to any decision made by the PC/SAC.

**Additional Comments On Reviewer Discussion:**

The discussions on this paper have been thorough and productive. Initially, the reviewers acknowledged the technical contribution but found it somewhat unclear. They also raised questions about the experiments. During the rebuttal stage, most of these concerns were addressed, although one reviewer still had unresolved questions about the baseline.

The positive reviewers expressed limited enthusiasm, while the negative reviewer indicated a willingness to reconsider the score. Overall, this paper remains borderline.

---

### Decision · Program_Chairs · 2025-01-22

Accept (Poster)